# Single-cell transcriptomic analysis of bloodstream *Trypanosoma brucei* reconstructs cell cycle progression and developmental quorum sensing

Emma M. Briggs [1,2✉], Federico Rojas [1], Richard McCulloch [2], Keith R. Matthews [1,3] & Thomas D. Otto [2,3]

Developmental steps in the trypanosome life-cycle involve transition between replicative and non-replicative forms specialised for survival in, and transmission between, mammalian and tsetse fly hosts. Here, using oligopeptide-induced differentiation in vitro, we model the progressive development of replicative 'slender' to transmissible 'stumpy' bloodstream form *Trypanosoma brucei* and capture the transcriptomes of 8,599 parasites using single cell transcriptomics (scRNA-seq). Using this framework, we detail the relative order of biological events during asynchronous development, profile dynamic gene expression patterns and identify putative regulators. We additionally map the cell cycle of proliferating parasites and position stumpy cell-cycle exit at early G1 before progression to a distinct G0 state. A null mutant for one transiently elevated developmental regulator, ZC3H20 is further analysed by scRNA-seq, identifying its point of failure in the developmental atlas. This approach provides a paradigm for the dissection of differentiation events in parasites, relevant to diverse transitions in pathogen biology.

[1] Institute for Immunology and Infection Research, School of Biological Sciences, University of Edinburgh, Edinburgh, UK. [2] Wellcome Centre for Integrative Parasitology, Institute of Infection, Immunity and Inflammation, University of Glasgow, Glasgow, UK. [3] These authors contributed equally: Keith R. Matthews, Thomas D. Otto. ✉email: emma.briggs@ed.ac.uk

African trypanosome parasites cause both human[1] and animal[2] trypanosomiases and are transmitted between hosts across sub-Saharan Africa by tsetse flies. During its life cycle *Trypanosoma brucei* undergoes several developmental transitions, comprising changes in nutrient-specific metabolism, morphology, organelle organisation and structure, and stage-specific surface protein expression[3], facilitating parasite survival and transmission. In the mammalian host, long slender bloodstream forms replicate extracellularly, increasing in numbers to trigger differentiation into short stumpy bloodstream- form parasites via a quorum sensing (QS) process[4,5], with ill-defined intermediate forms between these morphological extremes[6,7]. Stumpy forms remain arrested in the cell cycle[8] until ingested by a feeding tsetse fly, where they are pre-adapted to survive in the midgut[9,10]. Here, stumpy forms undergo a further differentiation event and re-enter the cell cycle as tsetse–midgut procyclic forms[9,11].

Slender and stumpy forms differ at both the transcript[12–17] and protein level[18,19], as do stumpy and procyclic parasites[15–17,19]. Reflecting their metabolism, slender forms show high levels of transcripts encoding glycosomal components (specialist organelles housing glycolytic enzymes)[9], whereas stumpy parasites upregulate transcripts related to a maturing mitochondrion as they prepare for the tsetse midgut. This allows for the metabolism of pyruvate, as well as proline and threonine, to generate ATP in low glucose conditions[9,13–15,20]. Consistent with exit from the cell cycle, stumpy parasites downregulate histone, DNA replication/repair, translation and cytoskeleton-related transcripts[15]. In addition, PAD (proteins associated with differentiation) transcripts are upregulated in stumpy forms and are required for further development into procyclics[21]. Transcripts encoding EP and GPEET repeat procyclin surface proteins expressed in tsetse–midgut forms are also elevated in stumpy forms, whereas variant surface glycoprotein (*VSG*s) transcripts, required for immune evasion by the parasite in the mammal, are reduced. Transcript analysis of *T. brucei* parasites isolated during parasitemia in vivo suggests some of these changes occur in early differentiating parasites, before morphologically detectable stumpy forms dominate at the peak of parasitemia[22].

QS-based development between slender and stumpy forms has been recently characterised, identifying several factors involved in detecting the differentiation stimulus[23], signal propagation[24,25] and implementation of cellular changes[24,26–28]. Yet, understanding the detailed developmental progression toward stumpy cells has been hampered by the asynchrony of this differentiation step, as has the relationship of regulatory genes to the various biological events of differentiation. Single-cell RNA sequencing (scRNA-seq) offers the opportunity to address this knowledge gap by studying individual cells in a heterogeneous population and thus identifying rare cell types and deciphering complex and transient developmental processes[29–31]. Recently, scRNA-seq has been used to study antigenic variation in *T. brucei*[32], as well as to describe the diversity of parasites in the tsetse fly salivary gland[33]. The latter study revealed early and late stages of metacyclic development, previously indistinguishable by population-based RNA-seq[34], highlighting the differing mRNA expression of surface proteins within the developing population[33].

Here, we apply scRNA-seq to analyse 8599 differentiating parasites progressing from bloodstream slender, through intermediate, to stumpy parasites in vitro using oligopeptide-rich bovine brain heart infusion (BHI) broth[23], deriving a temporal map of this transition at the transcript level based on individual cells. As no molecular markers of intermediate stages between the two developmental extremes are defined, detecting transcriptomic changes during this transition was previously not possible. By using scRNA-seq, we now provide a cell atlas detailing the

transcript level changes during development, revealing the absence of a discrete intermediate transcriptome, highlighting instead that only a limited number of transcripts are specifically enriched during the transition between slender- and stumpy forms. We further map the relative timing of biological events during differentiation, including exit from the cell cycle specifically prior to late G1, and identify novel genes regulated during the transition. Moreover, scRNA-seq analysis of a null mutant for one important regulator elevated during the slender to stumpy transition, ZC3H20[26,27], precisely maps where development fails in its absence in molecular terms. In combination, this provides a paradigm for the temporal mapping of developmental events and regulators during the parasite's dynamic differentiation programme in its mammalian host. This approach can be employed to study developmental regulators across species, providing a valuable strategy for deconstructing complex biological processes.

## Results

**scRNA-seq distinguishes slender and stumpy form transcriptomes**. To model stumpy development in vitro, pleomorphic *T. brucei* EATRO 1125 AnTa1.1 90:13 slender parasites were treated with oligopeptide-rich BHI broth, able to induce *T. brucei* bloodstream form differentiation in a titratable manner[23]. In the presence of 10% BHI, parasites underwent growth arrest (Fig. S1a), morphological change (Fig. S5), increased expression of the stumpy marker protein PAD1[35] (Fig. S1b), and increased the percentage of parasites containing one copy of the nucleus and one copy of the kinetoplast network (1N1K), indicating cell-cycle accumulation in G1/G0 and differentiation into stumpy forms[8] (Fig. S1c). After 72 h, 72.5% of cells expressed PAD1 (Fig. S1b) and 89.3% were in the 1N1K cell-cycle configuration (Fig. S1c). These data highlight that every time point contains a mix of parasites in terms of cell cycle and developmental stage, matching parasitaemias during an infection, such that bulk analysis of each sample would generate average gene expression of the heterogenous population. Therefore, to capture the individual transcriptomes of slender, differentiating and stumpy *T. brucei*, we combined parasites after 0, 24, 48 or 72 h of 10% BHI treatment in equal numbers. 15,000 cells of this heterogenous pool were then subjected to scRNA-seq using the Chromium Single Cell 3′ workflow (10X Genomics) and Illumina sequencing[36]. Two biological replicates (WT 1 and WT 2) were generated from independent time-course experiments. To ensure we captured high-quality transcriptomes for *T. brucei* parasites, we explored several parameters. First, we included *Leishmania mexicana* parasites in both samples to calculate the interspecies doublet rates where *T. brucei* and *L. mexicana* have been captured as one cell (8.6% and 7.2%, respectively) and levels of cross-species contaminating free RNA (median of 2.1% and 0.56%, respectively). Second, we calculated the percentage of non-polyadenylated ribosomal RNA (rRNA) contamination (median 0.61% and 0.85%, respectively). We also determined the percentage of transcripts originating from the parasite's mitochondrial genome, the kinetoplastid DNA (kDNA) maxi circle (median 0.56% and 0.43%, respectively), as lysed cells show preferential loss of cytoplasmic transcripts, thereby enriching mitochondrial transcripts[37]. These metrics are included in Supplementary data 1 and show the Chromium platform is a successful method for scRNA capture of *T. brucei*, as previously documented[33]. After filtering to remove transcriptomes of poorer quality using these metrics, *L. mexicana* transcriptomes, free transcripts and likely intraspecies *T. brucei* doublets using total RNA counts per cell, 8599 total cells remained (5321 and 3278, respectively; Fig. S2 and Supplementary data 1). Thus, of the starting ~15,000 cells in each replicate experiment, after quality control filtering 35% and 22%

of cells remained for analysis, respectively. Transcripts of 8758 genes (including 85.7% of the protein-coding reference transcriptome) were captured in at least 5 cells in both replicate experiments (10 cells total) and were retained in the data set. Per cell, 1052 and 1445 median genes were detected in each replicate experiment, respectively. These data compare favourably with other scRNA-seq studies of *T. brucei* using Chromium (298 genes average per cell[33]) and SMART-seq2 (1572 average genes per cell[32]) technologies.

After selecting the top variable genes in the data set, cells from the two replicate experiments were integrated and UMAP (Uniform Manifold Approximation and Projection[38]) was used to visualise the relationship between individual *T. brucei* transcriptomes in low dimensional space, where variation between transcriptomes dictates the space between cells (Fig. 1a–c). Cells from WT 1 and WT 2 experiments overlapped (Fig. 1a), indicating the capture of reproducible cell types in each biological replicate. Slender- and stumpy-like cells were clearly identifiable by the expression of marker genes: slender-associated glycolytic genes *GAPDH* and *PYK1*[39]; and the stumpy markers, *PAD2*[21] and *EP1* procyclin[13] (Fig. 1b). To validate our data set we performed differential expression analysis between these slender- and stumpy-like cells separately for each individual replicate, and compared the results with population-based bulk transcriptomic analysis of slender and stumpy form parasites captured at peak and low parasitaemia in vivo, respectively[12] (Fig. S3) 398 differentially expressed genes were identified in each replicate; 343 of these were identified in both experiments. In the first replicate experiment, 292 (73.4%) of the genes were also seen in the bulk RNA-seq study (adjusted *p*-value < 0.05), and 298 (76.6%) in the second replicate experiment. Of the genes identified as common to the scRNA-seq and bulk RNA-seq data, there was significant correlation between the change in transcripts between stumpy and slender forms (WT 1, $R = 0.777$ and WT 2, $R = 0.756$; Fig. S3b).

Several iterations of clustering analysis were performed with a series of resolution thresholds to assess if slender and stumpy populations could be subdivided into further clusters. The results of these analyses and associated marker genes can be explored in detail using a cell atlas webtool (http://cellatlas.mvls.gla.ac.uk/TbruceiBSF/). Here we discuss the result of a high stringency iteration, resulting in four distinct clusters of transcriptionally similar cells (Fig. 1c and Supplementary data 2), each appearing in comparable proportions per replicate (Fig. 1d). Differential expression analysis of transcripts between the slender A, slender B, stumpy A and stumpy B clusters identified 516 marker genes (adjusted *p*-value < 0.05, log2FC > 0.25; Supplementary data 2); relative expression of the top markers is plotted in Fig. 1e and raw expression of the 3 top transcripts marked in Fig. 1f. Gene ontology (GO) term enrichment analysis revealed the association of each cluster's marker genes with distinct biological processes (Fig. 1g). Several terms relating to cell-cycle processes were enriched in slender A and slender B marker genes (organelle and cilium organisation, chromosome segregation, and cell division), as well as cellular differentiation due to the presence of ZC3H20[26,27]. The slender B cluster of cells was additionally associated with genes implicated in cell communication (Ras-related protein *Rab5A*[40] and thimet oligopeptidase). Stumpy A marker genes were associated with the TCA cycle (mitochondrial malate dehydrogenase, two 2-oxoglutarate dehydrogenase E1 component encoding genes, and succinyl-CoA ligase [GDP-forming] beta-chain[41]), oxidation-reduction process (dihydrolipoyl dehydrogenase[4,42–44] and glutamate dehydrogenase[45]), rRNA metabolism (splicing factor *TSR1*[46], nucleolar RNA-binding protein *NOPP44/46-1*[47–49] and Lupus LA protein homologue[50]) and cell differentiation (zinc-finger protein 2;

ZFP2[51]). GO term analysis of stumpy B marker genes was limited due to their small number but included post-transcriptional regulators of gene expression, due to the presence of *ZC3H11*, which is also involved in heat shock response[52], and citrate transport due to the presence of *PAD2*, which is involved in detecting the signal for differentiation to procyclic forms[21].

In combination, the quality of our recovered transcripts, and high level of agreement between the scRNA-seq data and bulk mRNA expression, and with current understanding of the slender and stumpy cell types, validate Chromium as a highly effective platform for scRNA capture of *T. brucei* parasites. Moreover, the analysis revealed distinct slender and stumpy clusters, with significant differences between each population. Interestingly, a distinct cluster representative of a discrete intermediate stage transcriptome between slender and stumpy forms was not evident in any of the clustering iterations performed.

**Trajectory analysis of slender to stumpy differentiation.** As clustering analysis highlighted the transition from slender and stumpy cells involved overlapping gene expression and GO term association, we conducted trajectory inference and pseudotime analysis to study gene expression changes during stumpy development in detail. Individual cells were re-plotted as a PHATE (Potential of Heat-diffusion for Affinity-based Transition Embedding) map (Fig. 2a–c), which captures the local and global structure of high-dimensional data to preserve the continual progression of developmental processes[53]. Here, slender A and slender B clusters remained clearly separate, whereas stumpy A and stumpy B showed more extensive overlap (Fig. 2a). A linear trajectory starting from slender A cells was identified (Fig. 2b). Slender and stumpy marker gene (*GAPDH*, *PYK1*, *PAD2* and *EP1*) expression across the trajectory confirmed capture of the transition from slender to stumpy forms (Fig. 2c). In total, 1791 genes were identified as differentially expressed as a function of pseudotime (*p*-value < 0.05, fold change > 2) and were grouped into 9 modules (A–I) of co-expressed genes that showed similar patterns of expression across differentiation (Fig. 2d and Supplementary data 2). Of these, 1222 genes (68.2%) were previously found to be significantly (adjusted *p*-value < 0.05) differentially expressed between slender and stumpy enriched populations of *T. brucei* isolated from low and peak parasitaemia in vivo[12], confirming the physiological relevance of oligopeptide-induced differentiation in vitro (Fig. 2e). Proportionally fewer genes in modules A (transiently downregulated) and E (transiently upregulated) had been identified in bulk RNA-seq data as differentially expressed (50% and 26.1%, respectively) compared to the remaining modules (61.3–77.5%), highlighting the ability of single-cell analyses to reveal transient events in an asynchronous developmental trajectory[12].

GO term enrichment for biological processes associated with each gene module revealed the relative order of biological events during slender to stumpy development (Fig. 2f). Processes upregulated at the start of the trajectory included chromosome segregation and regulation of cytokinesis (module B), and these were then followed by cell division (modules C and D), indicating the progression of the later stages of the cell cycle. Module C also included genes linked to signalling, including *PKA-R* (previously identified as a potential signal transducer during stumpy development in response to hydrolysable-cAMP[24]) and the known differentiation regulator *RBP7B*[24,25]. Module D genes, which are broadly expressed across the slender parasites, include genes associated with the cell cycle as well as several post-transcriptional regulators of gene expression, including *RBP9* and *RBP10*, which are both known to positivity regulate the bloodstream form cell type[54]. Module E consisted of transiently

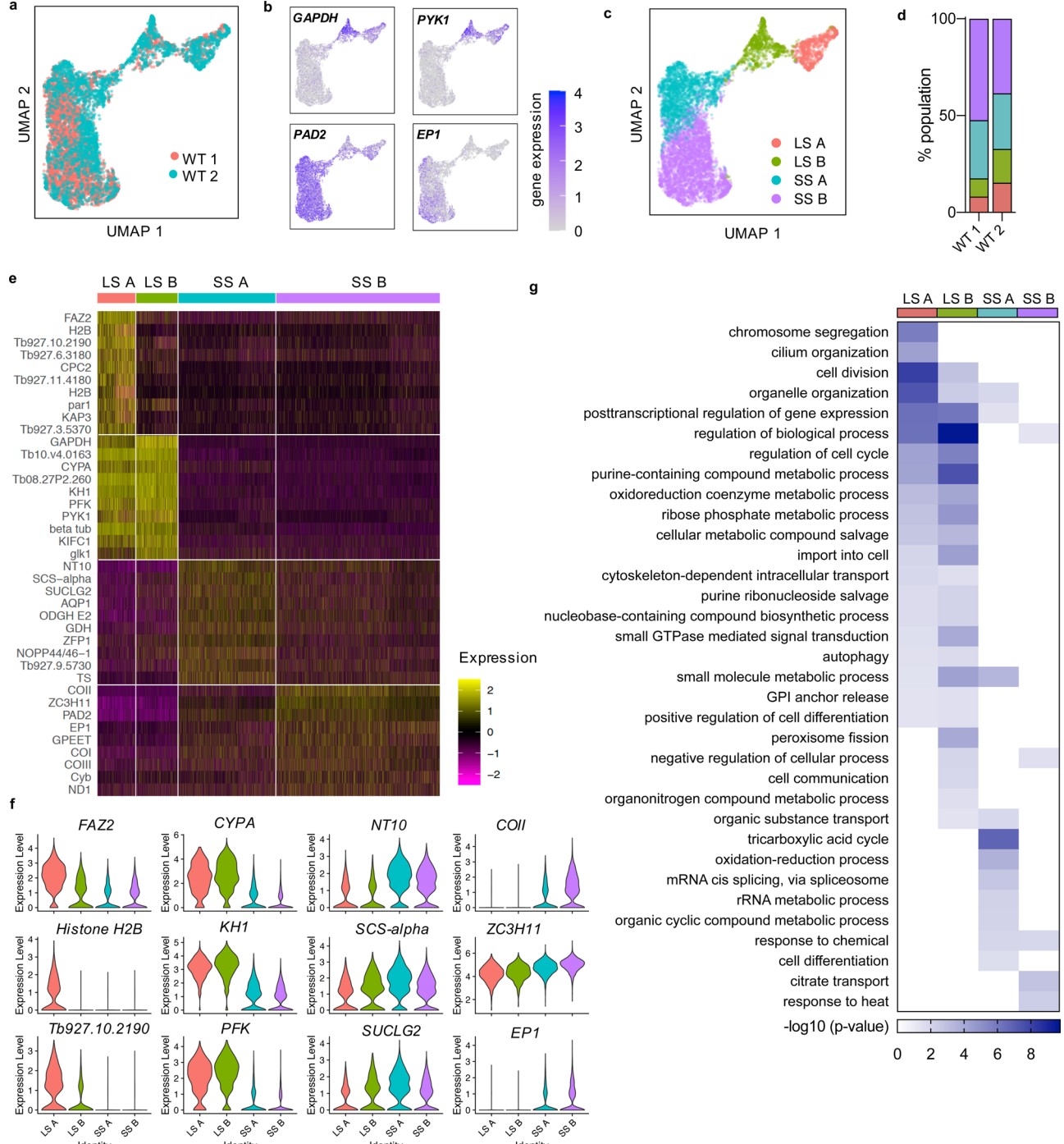

**Fig. 1 Sequencing of individual *T. brucei* transcriptomes during bloodstream differentiation in vitro. a** Low dimensional plot (UMAP) of each cell after filtering. Each point is the transcriptome of one cell positioned according to similarity with neighbouring transcriptomes, coloured by replicate experiment (WT 1 in red and WT 2 in blue). **b** UMAP of integrated WT parasite transcriptomes coloured by transcript counts for two slender marker genes (*GAPDH*; Tb927.6.4280 and *PYK1*; Tb927.10.14140) and two stumpy marker genes (*PAD2*; Tb927.7.5940 and *EP1*; Tb927.10.10260). Scale shows raw transcript count per cell. **c** UMAP of WT parasites from both replicates, coloured by cluster: Long slender (LS) A (red), LS B (green), short stumpy (SS) A (blue) and SS B (purple). **d** Percentage of parasites in each cluster for each replicate experiment. Colours as in (**c**). **e** Heatmap showing relative expression levels (log2 normalised z-score) of the top 10 maker genes of each cluster identified in (**c**). Each row is one gene coloured by relative expression. Where no gene name or symbol was available, the gene ID is shown. Each column is one cell grouped according to cluster. **f** Violin plots showing the expression of each of the top 3 marker genes per cell, divided by cluster. Y-axis shows the raw transcript count per cell. **g** Gene ontology (GO) enrichment for biological processes linked with maker genes for each cluster. Scale shows the −log10(*p*-value), calculated with two-tailed Fisher's exact test, for each term enrichmened per cluster.

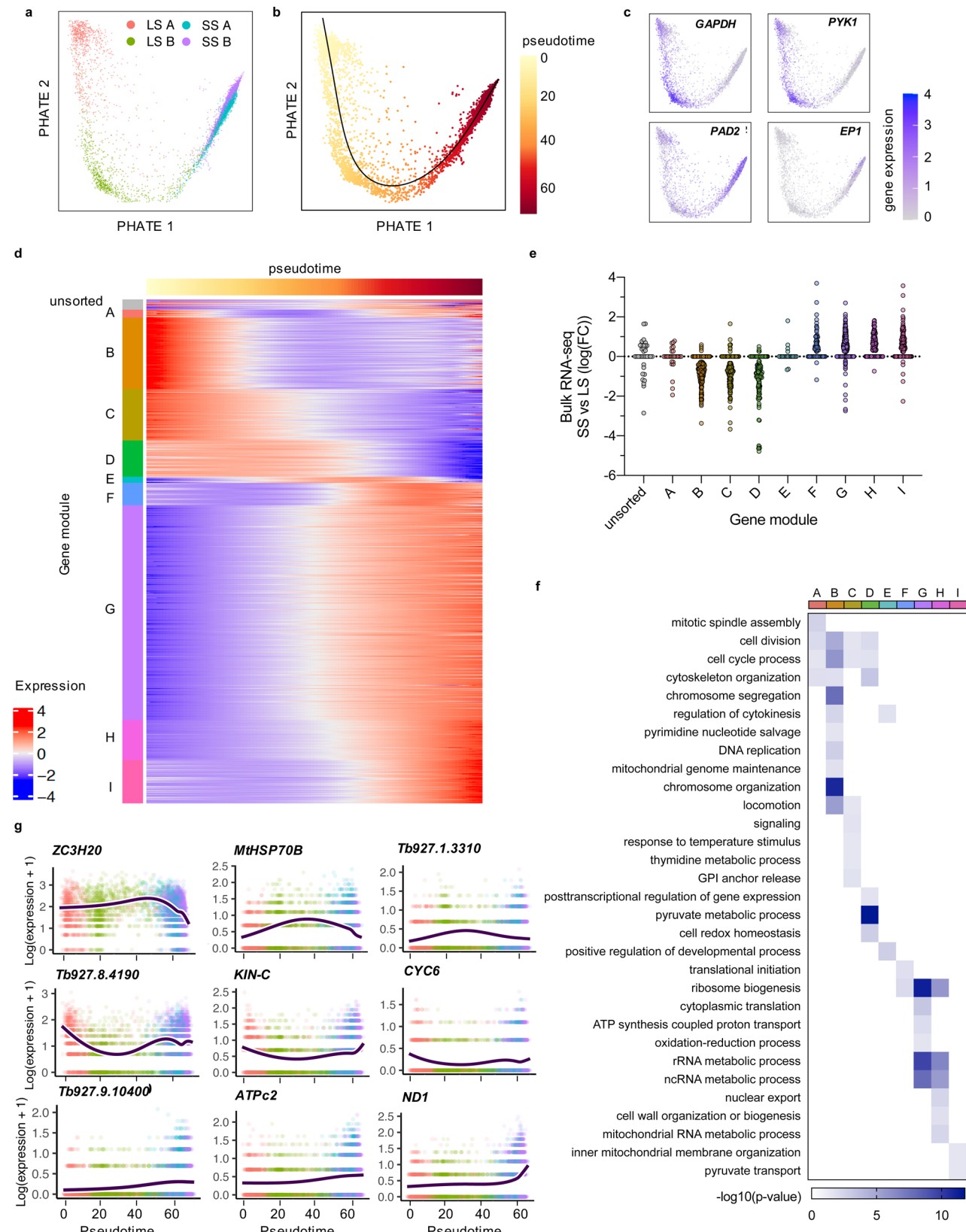

upregulated genes, including the known stumpy and procyclic developmental regulator *ZC3H20*[26,27,55]. Module F–I genes peaked in the later stages of development and include five genes identified in a reverse-genetic screen for stumpy development factors, including the chromatin regulator *ISWI* and hypothetical protein Tb927.11.1640[24]. The latter modules also include genes

relating to the electron transport chain (*ATPF1A*, *ATPB* and mitochondrial ATP synthase delta chain[41]), as well as kDNA-encoded genes (*RPS12*, *ND1*, *COI-III*, *NDH4* and *Cyb*). Beyond these annotated genes, 551 hypothetical genes were identified as differentially expressed during slender to stumpy differentiation, including Tb927.1.3310 (transiently upregulated), Tb927.8.4190

**Fig. 2 Pseudotime analysis reveals dynamic gene expression in slender to stumpy differentiation.** PHATE plots of individual parasite transcriptomes coloured by **a** cluster identity (long slender (LS) A in red, LS B in green, short stumpy (SS) A in blue and SS B in purple), **b** pseudotime values and **c** raw marker gene transcript count as in Fig. 1b. **d** Heatmap plot of relative expression levels (log2 normalised z-score) of 1791 genes with differential expression significantly associated with the trajectory (identify with associationTest; *p*-value > 0.05, FC > 2). Top track shows pseudotime as in (**b**). Genes are clustered by expression pattern over pseudotime into 9 modules of co-expressed genes, indicated to the left. Unsorted genes are indicated (grey). **e** Fold change of the 1791 differentiation-associated genes in bulk in vivo-derived RNA-seq data, comparing stumpy (peak parasitemia) and slender (low parasitemia) populations. Each point is one gene, grouped and coloured according to the co-expressed module identified in (**d**). **f** Biological progress gene ontology (GO) term analysis of differentiation-associated genes grouped by co-expressed module. Significance of selected GO terms enrichment in each co-expressed module of genes, indicated on the top track, is plotted. Scale shows the −log10(*p*-value), calculated with two-tailed Fisher's exact test, for each term enriched per module. **g** Gene expression (log2(transcript count +1)) across pseudotime from slender to stumpy differentiation of 9 genes identified as transiently upregulated (*ZC3H20*, *MtHSP70B*, *Tb927.1.3310*), transiently downregulated (*Tb927.8.4190*, *KIN-C*, *CYC6*) or not sorted into a gene expression module (*Tb927.9.10400*, *ATPc2*, *ND1*). Each point is one cell coloured by cluster as in (**a**). Dark blue line is smoothed average expression across pseudotime.

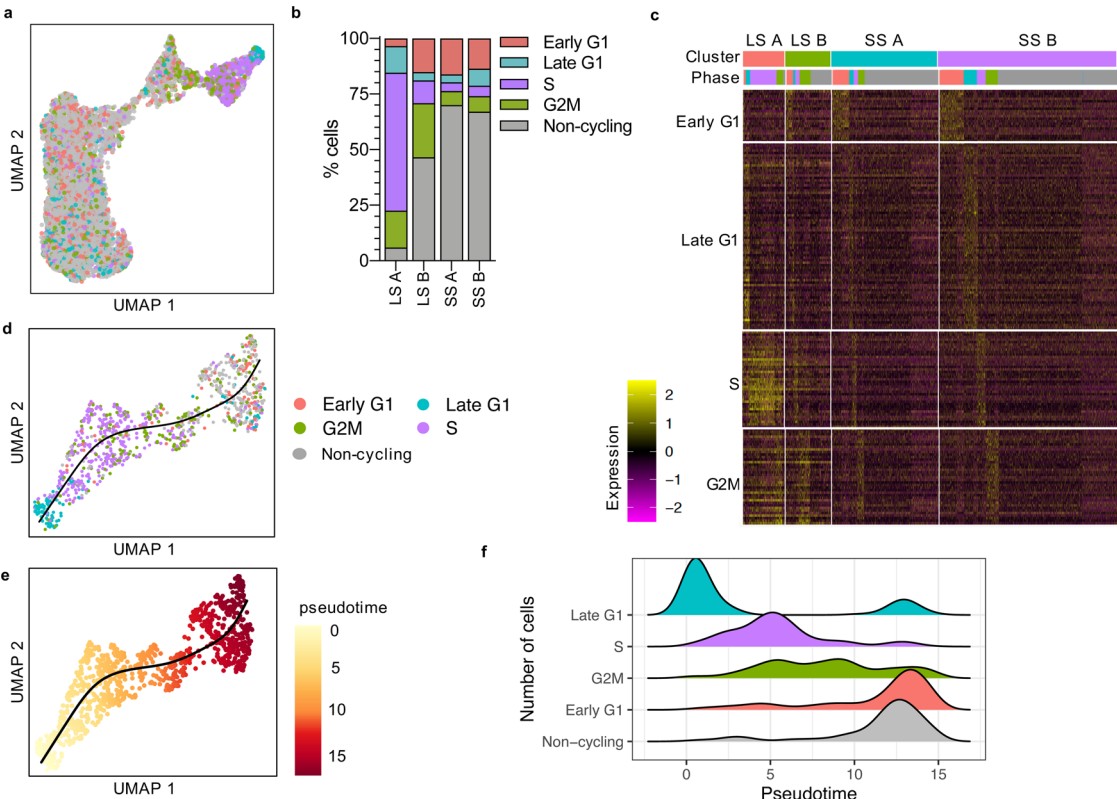

**Fig. 3 Differentiating *T. brucei* exit the cell cycle in early G1. a** UMAP of WT cells coloured by assigned cell-cycle phase. Key as in (**b**): Early G1 in red, Late G1 in blue, S in purple, G2M in green and non-cycling in grey. **b** Percentage of cells in each cell-cycle phase for each cluster. **c** Heatmap of relative expression levels (log2 normalised z-score) of cell-cycle phase marker genes (rows, grouped by phase). Each column is one cell grouped by cluster identity and cell-cycle phase. Phase key as in (**b**). **d** UMAP of slender A and slender B cells coloured by cell-cycle phase. Black line indicates inferred trajectory. **e** UMAP of slender A and slender B cells coloured by pseudotime value. Black line indicates inferred trajectory. **f** Ridgeplot showing the number of cells in each cell-cycle phase, or non-cycling, across pseudotime depicted in (**e**). Colours as in (**b**).

(transiently downregulated), and Tb927.9.10400, which peaks in stumpy cells (Fig. 2g).

Pseudotime analysis was able to identify novel genes differentially expressed during bloodstream form differentiation, as well as each gene's detailed expression pattern. The relative timing of events, from proliferation (chromosome segregation, cytokinesis), cell-cycle exit, cell remodelling, through to a maturing mitochondrion and expression of procyclin surface protein transcripts, can be inferred from these expression patterns. In addition, the expression peaks of known and putative developmental regulators were identified relative to this progression.

**Differentiating *T. brucei* exit the cell cycle in early G1.** To analyse coordinated cell-cycle exit and expression of stumpy-

associated transcripts, we assigned each cell to a cell-cycle phase using marker genes previously identified by bulk RNA-seq analysis[56] (Fig. 3a, b). Cells which did not overexpress any set of cell-cycle phase marker genes were labelled "non-cycling". Analysing the clusters presented in Fig. 1, revealed that 94.1% of slender A parasites were expressing cell-cycle marker genes, whereas slender B cells included 46.6% that were labelled as non-cycling. Interestingly, these slender non-cycling cells were positioned most proximal to stumpy cells (Fig. 3a). The percentage of non-cycling cells increased to 70.0% and 67.0% in stumpy A and stumpy B clusters, respectively (Fig. 3b). Although stumpy A and stumpy B cells were marked mainly as non-cycling, some cells were expressing cell-cycle phase marker genes (Fig. 3c), with the most common cycling cells being labelled as early G1 (Fig. 3b). Subsetting and replotting only slender A and slender B clusters

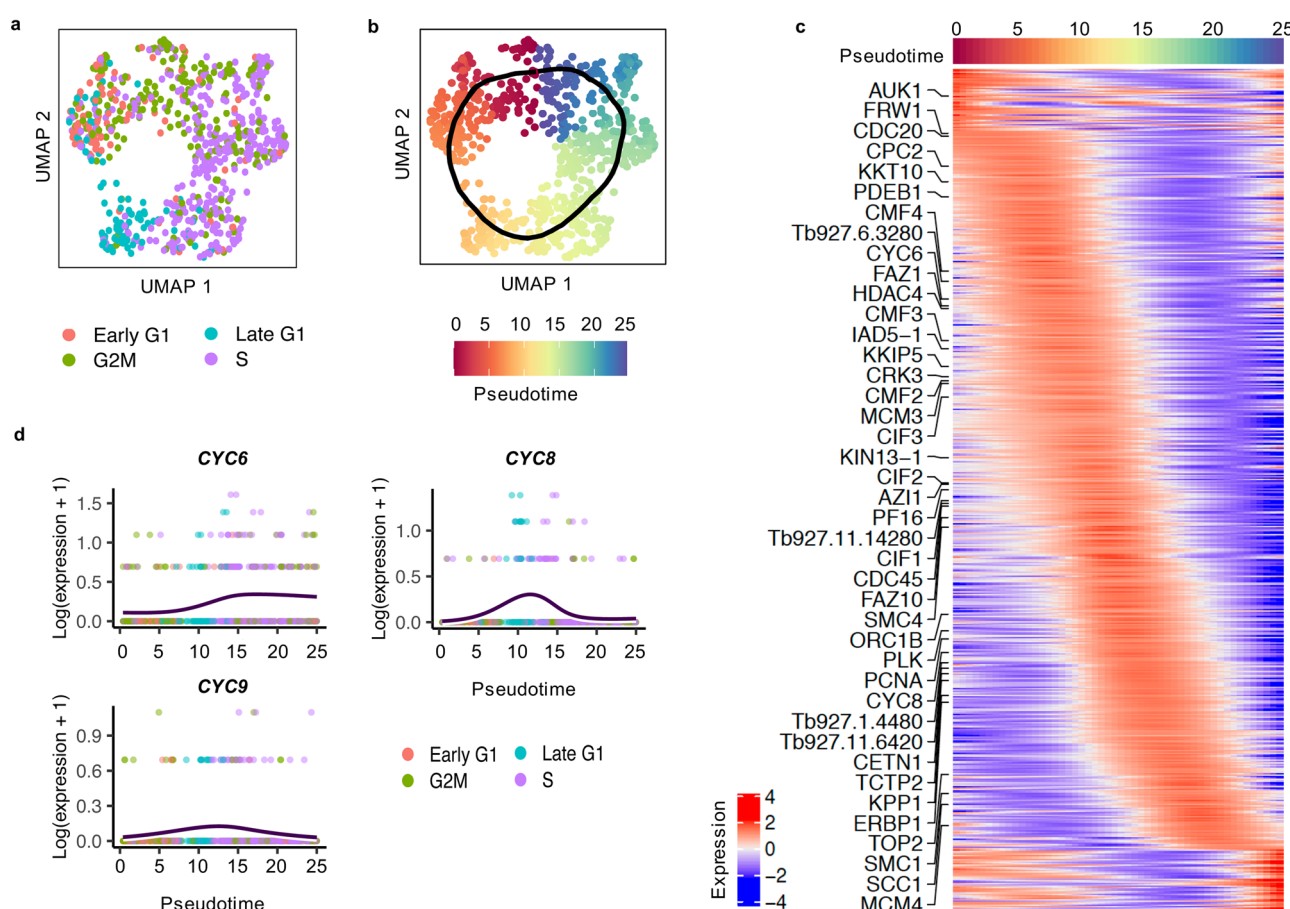

**Fig. 4 Identification of genes differentially expressed during the slender form cell cycle. a** UMAP of only cycling (as labelled in Fig. 3a) slender A and slender B cells re-plotted, using genes variable within the cycling slender population. Cells are coloured by cell-cycle phase: Early G1 in red, Late G1 in blue, S in purple, G2M in green. **b** UMAP of cycling slender cells coloured by assigned pseudotime value. The black line indicates the inferred circular trajectory. **c** Heatmap of relative expression levels (log2 normalised z-score) of genes significantly differentially expressed over the cell cycle (identify with associationTest; $p$-value > 0.05, FC > 2). Genes associated with the GO term "cell cycle process" are labelled. **d** Expression levels (y-axis; log2(expression + 1)) of 3 cyclin genes (*CYC6*, *CYC8* and *CYC9*) over cell-cycle pseudotime (x-axis). Each point is one cell coloured by cell-cycle phase (as in **a**) and dark blue line shows smoothed average expression over pseudotime.

revealed the progression of cells through the cell cycle and subsequent exit by some cells (Fig. 3d). Inferring a trajectory and pseudotime values (Fig. 3e) clearly showed most cells were in the late G1 phase at the beginning of pseudotime, followed by S phase and G2M phase (Fig. 3f). At the end of the trajectory most cells were in early G1 or were non-cycling. These plots reveal that during bloodstream form differentiation, the majority of cells exit the cell cycle specifically in early G1.

**Transcript abundance changes during the slender cell cycle.** As replicating slender bloodstream form cells were captured in these experiments, we next asked if the scRNA-seq data could reveal greater detail than currently available on gene expression changes during the cell cycle. After isolating slender A and slender B cells, the non-cycling cells were also removed before a UMAP was plotted (Fig. 4a). Cells formed a clear circle and were grouped by cell-cycle phase in the expected order. To assess gene expression changes during the cell cycle, we fitted a cyclical trajectory to this plot and assigned pseudotime values (Fig. 4b). Testing each gene for expression patterns associated with pseudotime identified 884 that were significantly (adj. $p$-value < 0.05) differentially expressed (Supplementary data 3). Of these, 612 changed in abundance by more than two-fold (Fig. 4c). GO term enrichment revealed expected GO terms associated with the cell cycle; 41 genes

associated with the term "cell cycle process" are highlighted in Fig. 3d. Amongst these 41 genes were several where protein levels or distribution have been shown to match the scRNA-seq predicted cell-cycle timing of expression, including *ORC1B*[57], *AUK1*[58], *PCNA*[59] and *KKIP5*[60]. Many further genes displayed cell-cycle-regulated expression that has not yet been explored (Supplementary data 3). For instance, three cyclin genes were identified, each with a distinct expression profile; *CYC6* increased during S phase, *CYC8* peaked sharply in late G1 immediately prior to S phase, and *CYC9* showed a similar but less pronounced increase in the same phase (Fig. 4d). All three expression patterns are in agreement with previously generated bulk RNA-seq analysis of cell-cycle sorted populations[56].

**ZC3H20 null parasites fail to differentiate in vitro.** The developmental analysis highlighted differential expression patterns of several stumpy development regulators, including ZC3H20, which peaks in expression at the slender B to stumpy transition in pseudotime (Fig. 2g). As ZC3H20 has previously been shown to be required for differentiation in vivo and in vitro[26,27], we repeated scRNA-seq analysis with a ZC3H20 null *T. brucei* line[27] to investigate where parasites fail in their development to stumpy forms with respect to transcriptome changes and, potentially, to identify direct or indirect mRNA targets of

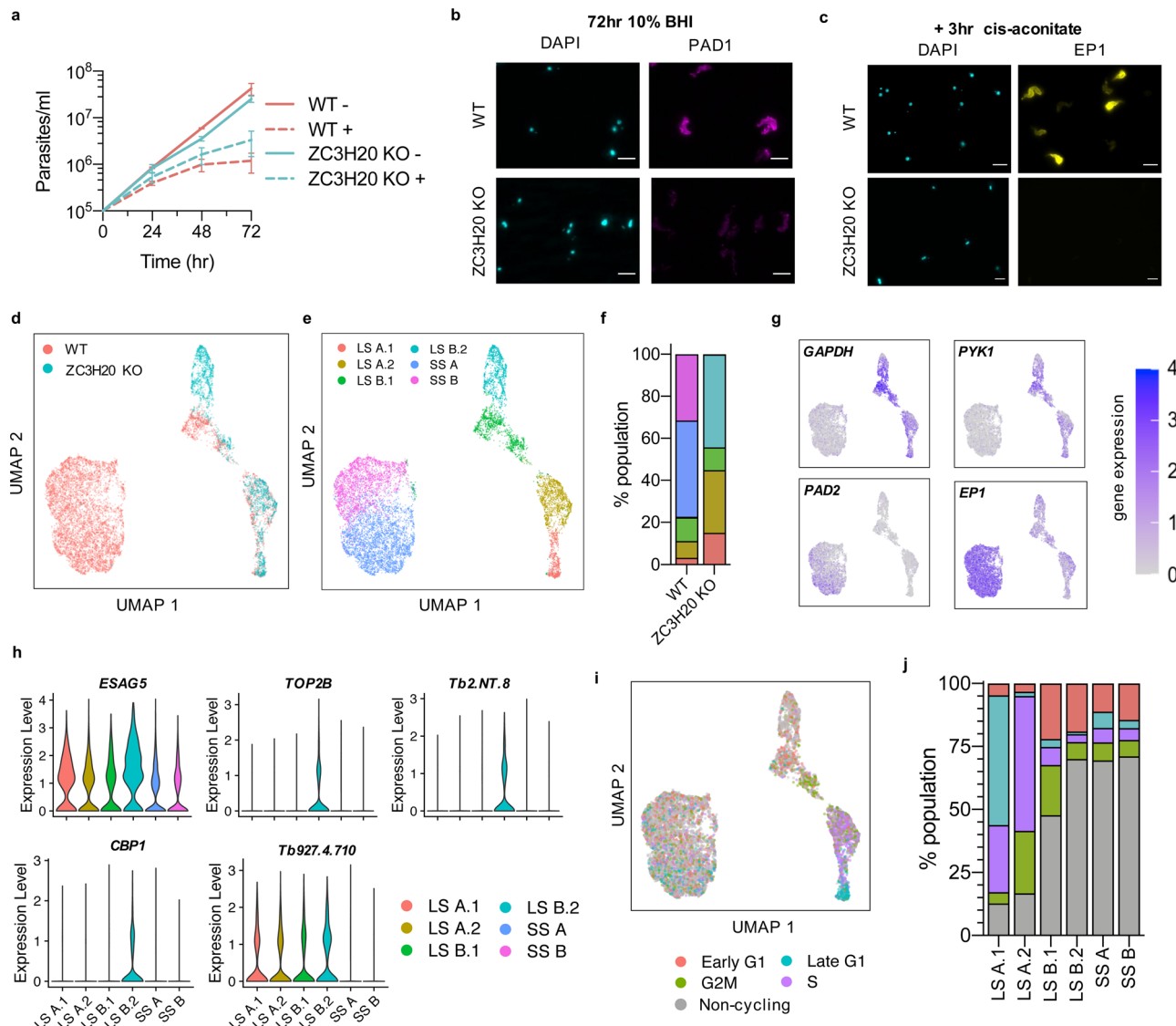

**Fig. 5 scRNA-seq analysis of differentiation incompetent ZC3H20 KO _T. brucei_ parasites. a** Cumulative growth of WT (red) and ZC3H20 KO (blue) _T. brucei_ in culture with (dashed line) and without (solid line) 10% BHI broth. _y_-axis shows parasites per ml on a log10 scale. Data are presented as the mean +/− SD of three independent replicates. **b** Staining of WT and ZC3H20 KO parasites with anti-PAD1 antibody after 72 h incubation with 10% BHI broth. Scale, 5 μm. Two biological replicates were performed. **c** EP1 staining of WT and ZC3H20 KO after 3 h treatment with _cis_-aconitate to induce differentiation of 72 h BHI+ samples into procyclic forms. Scale, 5 μm. Two biological replicates were performed. **d** UMAP plot of integrated WT cells (red) and ZC3H20 KO cells (blue) coloured by cell type. **e** UMAP plot of integrated WT and ZC3H20 KO cells coloured by cluster identification (long slender (SL) A.1 in red, LS A.2 in yellow, LS B.1 in green, LS B.2 in aqua, shorty stumpy (SS) A in dark blue and SS B in purple. **f** Proportion of cells in each cluster identified in integrated WT and ZC3H20 KO cells. Colours as in (**e**). **g** UMAP of WT and ZC3H20 KO parasites coloured by transcript count of marker genes, as in Fig. 1b. **h** Violin plots of the top long slender B.2 marker gene expression in each cluster identified in (**e**). **i** UMAP of integrated WT and ZC3H20 KO cells coloured by cell-cycle phase (Early G1 in red, Late G1 in blue, S in purple, G2M in green and non-cycling in grey). **j** Percentage of cells in each cell cycle phase, or non-cycling, divided by cluster. Colours as in (**i**).

ZC3H20 itself. We first tested the effect of 10% BHI broth on ZC3H20 null _T. brucei_ parasites (ZC3H20 KO)[27] (Fig. 5a–c). In the presence of 10% BHI broth, replication of ZC3H20 KO parasites slowed after 72 h of culture (Fig. 5a). In addition, the ZC3H20 KO parasites failed to express the stumpy marker protein PAD1 (Fig. 5b) and did not change in cell width, indicating they did not progress towards a typical stumpy morphology (Fig. S5). To assess if ZC3H20 KO parasites are irreversibly growth arrested in response to BHI, ZC3H20 KO and the parental cell line were treated with fresh BHI for 48 h before washing the parasites and returning them to BHI-free media. The growth of ZC3H20 KO parasites recovered rapidly once returned to BHI-free media, whereas the wild-type parasites showed a delayed

return to growth after 48 h of recovery (Fig. S4). In addition, we tested the ability of WT and ZC3H20 KO parasites exposed to BHI for 72 h to differentiate into procyclic cells. Consistent with their inability to generate stumpy forms, after 3 h of _cis_-aconitate treatment and incubation at 27 °C, none of ZC3H20 KO parasites expressed EP1 procyclin, in contrast with 84.34% of WT parasites (mean of two biological replicate experiments), confirming ZC3H20 KO _T. brucei_ fail to differentiate into functional stumpy cells when exposed to BHI broth (Fig. 5c).

ZC3H20 KO parasites were next cultured in 10% BHI for 0, 24, 48 or 72 h and subjected to scRNA-seq, as for WT samples. After quality control filtering, 2295 cells (median 1051 genes per cell) remained and were integrated with the WT cells before

dimensional reduction was performed and the results were plotted as UMAPs (Fig. 5d, e, g). Clustering the ZC3H20 KO and WT integrated cells resulted in six distinct clusters: stumpy A and stumpy B, and four slender clusters, slender A.1, slender A.2, slender B.1, slender B.2 (Fig. 5f). These were identified as slender- and stumpy-like cells by the expression of marker genes *GAPDH*, *PYK1*, *PAD2* and *EP1* (Fig. 5g). Whereas 77.3% of WT cells were found in clusters stumpy A or stumpy B, only 0.13% of ZC3H20 KO cells were in either, consistent with near-complete ablation of stumpy formation in the mutant parasites (Fig. 5f). The majority of WT slender parasites grouped as members of the slender A.2 and slender B.1 clusters (8% and 11.1% of all parasites, respectively), whereas ZC3H20 KO cells were divided between the four slender clusters. Notably, the slender B.2 cohort was comprised almost entirely of ZC3H20 KO parasites (comprising 44.2% of total ZC3H20 KO vs 0.4% of total WT cells). Marker gene analysis between clusters (Fig. S6 and Supplementary data 4), identified 97 marker genes upregulated in slender B.2 cells. Of these, just 22 genes were significantly upregulated in cluster slender B.2 alone, the top 5 genes being expression site-associated gene 5 (*ESAG5*), DNA topoisomerase II beta (*TOP2B*), one non-coding RNA gene (Tb2.NT.8), Family S10 protein *CBP1*, and a hypothetical gene Tb927.4.710 (Fig. 5h). Cell-cycle status contributed considerably to the clustering analysis of WT and ZC3H20 KO integrated cells, as slender cells clearly grouped by cell-cycle phase (Fig. 5i). Interestingly, cluster slender B.2 is enriched for non-cycling cells to the same level as stumpy clusters, consistent with the reduction in the growth of ZC3H20 KO cells in the presence of BHI (Fig. 5j).

**Transcript down- and upregulation occur independently**. To compare the transcriptomic changes in ZC3H20 KO and WT *T. brucei* after BHI treatment in more detail, we inferred a trajectory from the WT and ZC3H20 KO integrated parasites (Fig. 6a). Doing so identified a branched trajectory: early in pseudotime, WT and ZC3H20 KO parasites were transcriptionally similar and arranged on the same lineage; later, there was a clear branch in their comparative development, ending for WT in stumpy cells and in slender B.2 for ZC3H20 KO cells (Fig. 6b). To understand this change, we first assessed the expression of differentiation-associated genes identified previously in WT parasites (Fig. 2d), mapping them across the truncated trajectory branch of the ZC3H20 KO cells (Fig. 6b, c). Of the 1791 genes identified as differentially expressed during stumpy development in WT cells, 494 genes significantly changed in expression in ZC3H20 KO parasites across the truncated trajectory (Fig. 6c). Of these, the majority (86.03%) were less highly associated with the ZC3H20 KO trajectory relative to that of the WT cells (Fig. 6c and Supplementary data 4). 81.2% of these genes were part of co-expression modules B–D, which decreased in expression during stumpy development in WT parasites (Fig. 6d). These included genes involved with glycolysis and the mitotic cell-cycle (Fig. 6e). Genes differentially expressed in the ZC3H20 KO trajectory and belonging to expression modules E–H included heat shock 70 kDa protein mitochondrial precursor subunits B and C, and three components of the TCA cycle, of which only *2-OGDH E1* increased to a similar level in ZC3H20 KO and WT trajectories (Fig. 6e). Hence, ZC3H20 KO cells downregulated transcripts associated with slender cells when exposed to the BHI differentiation stimulus, matching the response of WT cells. However, ZC3H20 KO parasites failed to upregulate transcripts later in development that are required for stumpy formation, and this point of dysregulation coincided with the peak of ZC3H20 expression during normal WT differentiation (Fig. 2g).

To identify regulators of early stumpy development, we looked for genes which changed significantly in abundance from the start of the trajectory to a point just downstream of the ZC3H20 KO branch (yellow dots, Fig. 6a). Two hundred and forty-four genes changed in transcript abundance between these points and were associated with trajectory progression (Fig. 6f and Supplementary data 4). One hundred of these genes were associated with both WT and ZC3H20 KO trajectories (*p*-value < 0.05) and included cell-cycle-associated genes, as expected. *PAD2*[35] was associated with both trajectories but showed different patterns of expression (Fig. 6g). One hundred and nine genes were differentially expressed only early in the trajectory of WT parasites, including 32 encoding hypothetical proteins (Supplementary data 4). Of these hypothetical genes, Tb927.6.4270 has been shown to decrease mRNA stability[61]. Given that ZC3H20 KO cells fail to exit the cell cycle and differentiate to stumpy forms (Figs. 5 and S4), the genes altered in WT cells only are expected to include regulators of commitment to cell-cycle exit and differentiation. In addition to the hypothetical genes, these included retrotransposon hotspot protein 4 (*RHS4*), heat shock response protein *HSP70.b*, zinc-finger protein *ZC3H29*, putative mRNA transport regulator *MTR2*, and six genes linked to ribosomal RNA biogenesis, including *NMD3* (Fig. 6g).

In summary, comparing the differentiation of WT and differentiation incompetent ZC3H20 KO cells through scRNA-seq has allowed the identification of (a) the failure point of ZC3H20 KO cells during the temporal profile of differentiation, (b) putative early regulators of differentiation and, potentially, (c) the direct and indirect targets of ZC3H20 altered specifically during differentiation.

## Discussion

Although extensively studied, *T. brucei* differentiation from slender to stumpy bloodstream forms has remained difficult to dissect in detail due to the asynchronous nature of this life-cycle transition. Here, we used oligopeptide induction of differentiation[23] in combination with scRNA-seq to deconvolve this process at the transcript level. This dissection would not be possible using bulk RNA-seq analysis which reports only the average of the asynchronous and heterogeneous differentiating populations (Figs. S1 and S5, and ref. [23]). Moreover, enrichment methods based on sorting are precluded by the absence of molecular markers or a defined morphology for intermediate stages (Fig. S5). The scRNA-seq approach used here revealed several details of this process, summarised in Fig. 6. These included: the lack of a discrete intermediate transcriptome; the precise timing of cell-cycle exit, immediately prior to late G1; the transient expression of several genes not identified by bulk analysis; and the expression timing of known and putative differentiation factors during the developmental processes. Using scRNA-seq to study ZC3H20 KO parasites, we were also able to validate the essentiality of ZC3H20 for differentiation and position its action specifically at the major slender to stumpy transition point where *ZC3H20* transcripts peak in abundance. Finally, by comparing the developmental progressions of WT and ZC3H20 KO parasites, we revealed putative regulators of commitment. An additional analysis of proliferative slender cells provided detailed gene expression patterns of both known and novel cell-cycle-regulated genes during the bloodstream form cell cycle.

Clustering differentiating WT *T. brucei* into groups of transcriptionally similar cells clearly identified two primary groups corresponding to slender and stumpy parasites, each of which could be further classified into two sub-slender and sub-stumpy clusters (Fig. 1). The transcript differences between clusters were mainly due to the cell-cycle phase of slender cells and the stage of

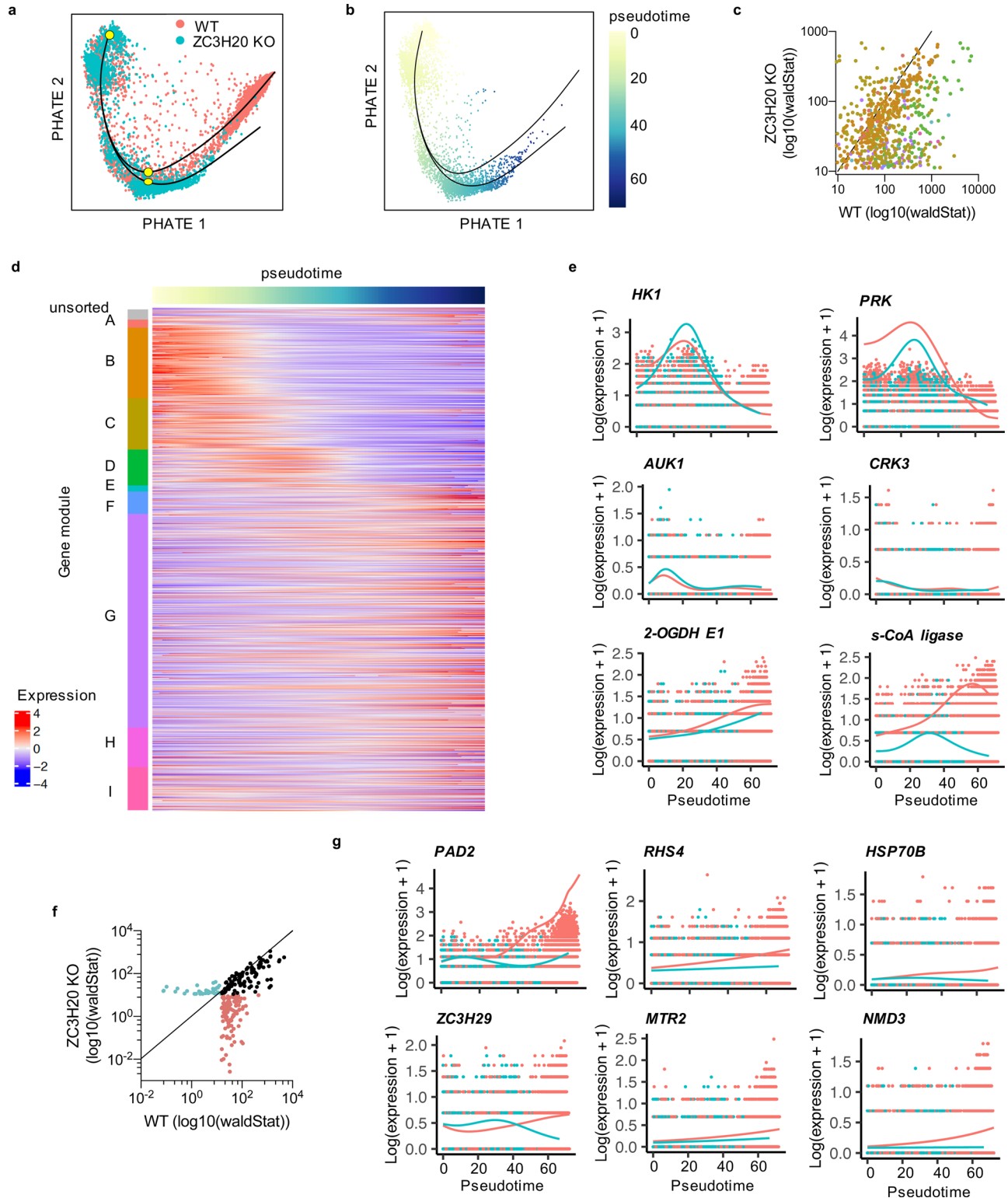

progression towards stumpy development (Figs. 1 and 2), with gene expression changes highlighting a progressive transition to stumpy forms with relatively few genes transiently changing in abundance during this differentiation process ($n = 51$, FC > 2). Thus, although discrimination of parasites between the extremes of the slender and slender morphotypes is possible microscopically, scRNA-seq analysis does not provide evidence for an intermediate form defined by the expression of a unique set of upregulated transcripts distinct from those in slender or stumpy

forms. Rather, cells expressing stumpy-associated transcripts appear to emerge directly from the G1 phase of replicative slender cells (Fig. 6a, see below). It remains posssible that an intermediate form could be defined by the expression of just a few distinct transcripts, or by changes in protein abundance modulated by translation- or protein-level regulation; such possibilities require further experiments to be tested.

Previous bulk transcriptomics identified marker genes of the *T. brucei* cell-cycle phases (early and late G1, S phase and G2/

**Fig. 6 Comparison of differentially expressed genes during differentiation of WT cells and differentiation incompetent ZC3H20 KO cells. a** PHATE map of WT (red) and ZC3H20 KO (blue) parasites. The black lines indicates branched trajectories. Yellow dots indicate points of analysis for early differentially expressed genes. **b** PHATE map of ZC3H20 cells only, coloured by pseudotime values assigned for the second lineage of the branch trajectory, black line. **c** Scatter plot of differentiation-associated genes also found to be differentially expressed in the ZC3H20 KO trajectory. Axes show the association score (log10(wald stat)) for each gene with the WT differentiation trajectory (x-axis) and ZC3H20 KO trajectory (y-axis). Each gene is coloured by is co-expression module identified in Fig. 2d. Black line indicates $x = y$. **d** Heatmap of the 1791 differentiation-associated genes identified in Fig. 2d. Relative expression levels (log2 normalised z-score) across the ZC3H20 KO trajectory is plotted for each gene, grouped by co-expression module. **e** Expression of example genes across the WT differentiation trajectory (red) and ZC3H20 KO trajectory (blue). X-axis; pseudotime, y-axis; log2(expression + 1). **f** Scatter plot of genes identified as early differentially expressed across the branched trajectory (between yellow points in **a**). Genes were identified by earlyDETest (p-value < 0.05 and FC > 2). Axes show the association score (log10(wald stat)) for each gene with the WT differentiation trajectory (x-axis) and ZC3H20 KO trajectory (y-axis). Genes are coloured by their significant association with the WT (red), ZC3H20 KO (blue), or both trajectories (black). **g** Expression patterns of examples of early differentially expressed genes as in (**e**).

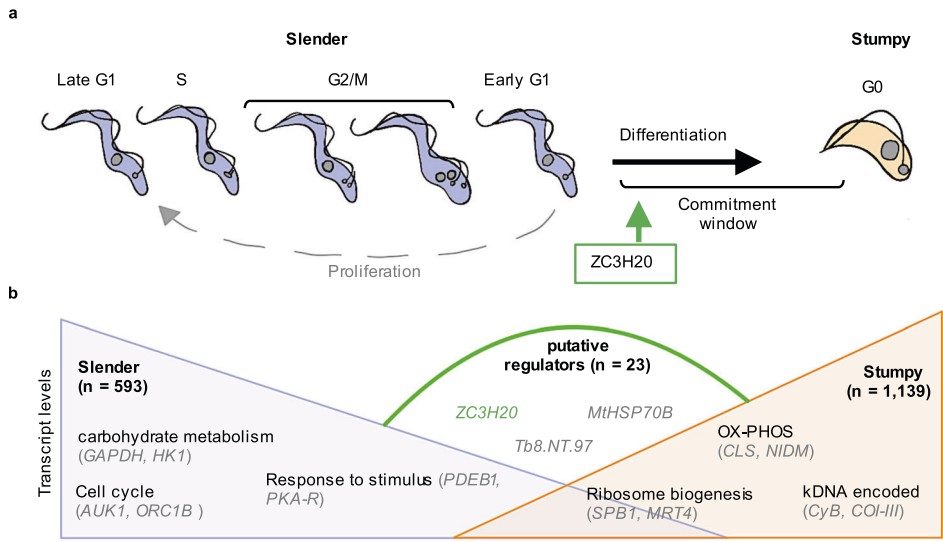

**Fig. 7 Model of the slender to stumpy transition in bloodstream trypanosomes. a** Slender forms (purple) complete the cell cycle, replicating and dividing first the kDNA mitochondrial genome (small grey circles) followed by the nuclear genome (large grey circles), replicating organelles (not shown) and the flagellum, completing cytokinesis (not shown) and finally reaching early G1. Here cells either pass to late G1 to re-enter the cell cycle (proliferation route; grey dashed line) or exit the cell cycle into a G0 phase and differentiate to stumpy forms (orange). The point of action for the essential regulator, ZC3H20, is indicated by the green arrow. Commitment to cell-cycle exit, morphology change and differentiation is likely to be controlled either by ZC3H20 or downstream factors; indicated by the "commitment window". **b** Slender-associated transcripts (purple) decrease during differentiation mostly before stumpy-associated transcripts increase (orange). Selected biological processes and example genes are indicated for each, as well as genes encoded on the kDNA maxi circle towards the end of differentiation. A small group of 23 genes show transiently increased transcript levels spanning the slender to stumpy transition point (green arch). These include the validated regulator ZC3H20, mitochondrial heat shock protein 70B (MtHSP70B) and non-coding RNA Tb8.NT.97. Thirty-six genes which were not sorted into co-expressed modules and 28 transiently downregulated genes are not shown. Full data set is included in Supplementary data 2. Diagrams are not to scale.

mitosis)[56], allowing us to define the most likely cell-cycle position of each cell in our scRNA-seq data set, including likely non-cycling G0 cells (Fig. 3). Defining a trajectory of slender cells moving through and exiting the cell-cycle, showed that late G1 stage cells are positioned at the start of the trajectory consistent with a cell-cycle receptive window[62], followed by S and G2/M phase cells. Early G1 and non-cycling cells were enriched closer to stumpy clusters, and stumpy clusters showed enrichment for non-cycling cells (Fig. 3). Hence, the major switch in transcriptome from slender to stumpy occurs during G1 and, specifically, before cycling cells re-enter late G1 (Fig. 7). Cells at other stages may be committed to differentiation but not yet arrested, as predicted by modelling[63]. The characteristics of this G0 phase await further exploration but provides an evolutionary divergent and tractable model for studying the conservation of quiescence signalling pathways, which are critical in many eukaryotic developmental processes[64–68]. Interestingly, some cells expressing stumpy-associated transcripts also expressed genes of an active cell-cycle phase, particularly early G1 (Fig. 3c). This may

reflect flexibility in the coordinated expression changes of differentiation and cell-cycle-associated genes, but may also be explained by relative dynamics of mRNA abundance for cell-cycle and developmental markers, or stochasticity.

We were additionally able to investigate the changes in transcript abundance during the proliferative slender cell cycle. Pseudotime analysis allowed us to profile the dynamic patterns of 884 genes found to be differentially expressed (Fig. 4), 386 of which had been previously identified in bulk RNA-seq analysis of synchronised procyclic T. brucei[56]. For example, we found CYC6, CYC8 and CYC9 peak at distinct points: CYC6[69–71] is known to regulate nuclear division and increased during S phase, persisting through G2M; predicted, but untested, mitotic cyclin CYC8 peaks very precisely in late G1/S-phase cells; and CYC9, which may have an indirect role in regulating cytokinesis[72], shows a similar but less pronounced increase in late G1/S. Having not undergone either selectional or chemical synchronisation procedures[56,73,74], this scRNA-seq derived cell-cycle atlas provides a relatively unperturbed picture of cell-cycle-regulated events in greater detail

than previously available, and suggests candidates for functional analysis. Distinctions between developmentally competent (pleomorphic) slender forms and adapted monomorphic forms used in previous studies may also be identified.

Trajectory inference and differential expression analysis of the slender to stumpy transition revealed the relative order of events during differentiation, information previously inaccessible due to the asynchronous progress of differentiation. Initially, there was higher abundance of transcripts linked to proliferation (Figs. 1 and 2), with cells completing the later stages of the cell cycle during the early stages of the differentiation trajectory, consistent with phase scoring analysis (Figs. 2 and 3). Thereafter, metabolic changes and activation of the mitochondrion occurred as expected[12,39]. Finally, expression of several kDNA-encoded genes (cytochrome oxidase subunits I–III and cytochrome B) and procyclin surface protein-encoding genes (EP1, EP2 and GPEET) was observed, reflecting preparation for differentiation to procyclic forms[12,15,17]. These changes correlated well with bulk mRNA analysis of in vivo-derived parasites[12], validating the use of BHI as an in vitro model of stumpy development. Transient expression patterns of several genes, not discernible in bulk RNA-seq and proteomic studies, were also observed. These included several hypothetical genes (Supplementary data 2), which may prove to be negative or positive regulators of differentiation.

Transiently upregulated genes also included the known differentiation regulator ZC3H20[26,27,55], confirming that we were able to identify developmental regulators via their gene expression patterns. We postulated that scRNA-seq analysis may enable us to map cells with diverse differentiation phenotypes onto our trajectory of WT differentiation, to assess the point at which genetically perturbed parasites fail to develop. We therefore exposed ZC3H20 KO parasites[27] to oligopeptides and confirmed that they fail to develop stumpy forms (Figs. 5a–c and S5). Trajectory inference revealed that ZC3H20 KO cells downregulate transcripts also downregulated in oligopeptide-stimulated WT parasites, including several glycolytic components, cell-cycle-regulated genes, and post-transcriptional regulators of gene expression (Fig. 6). This downregulation may contribute to the reduced growth of ZC3H20 KO parasites when exposed to BHI. Interestingly, there was a clear distinction between the downregulation of slender transcripts and the increase of stumpy-associated transcripts, which ZC3H20 KO parasites failed to upregulate to WT levels, including 12/19 ZC3H20 regulated mRNAs[26]. This finding suggests that ZC3H20 KO parasites perceive the differentiation signal and undergo early steps of differentiation, but do not commit to cell-cycle exit and further development to stumpy forms. The transcriptional differences between the mutant and WT parasites, which included 28 hypothetical genes, four of which are known to regulate mRNA stability[61], identifies putative 'early' regulators of commitment (Fig. 6f and Supplementary data 4), which should be prioritised for experimental follow-up.

In conclusion, our data demonstrate that transcript level changes in parasites can be used to compile maps of both the cellcycle and the asynchronous slender to stumpy differentiation process. Although lowly expressed genes may have been missed due to the low sensitivity of scRNA-seq approaches, these data can still be mined to identify regulatory genes of individual events that make up each process. We further characterised mutant parasites by the same approach, positioning the site of action of one regulator (ZC3H20) in the developmental time course. If iterated for different genes, this method can be exploited to derive hierarchies of gene action during differentiation in this and other life-cycle stages, species and development processes.

## Methods

**Trypanosoma brucei cell lines and culture.** Trypanosoma brucei EATRO 1125 AnTat1.1 90:13 parasites[75] were used as pleomorphic wild type (WT) in all experiments. The ZC3H20 KO null parasites were previously generated in the EATRO 1125 AnTat1.1 cell line transfected with plasmid pJ1399 (gifted by Dr. Jack Sunter), containing T7 polymerase and CRISPR/cas9, by replacement of both alleles of Tb927.7.2660 with blasticidin S deaminase[27]. All parasites were grown free from selective drugs in HMI-9 medium[76] (Life Technologies), supplemented with 10% foetal calf serum at 37 °C, 5% $CO_2$. For induction of differentiation, parasites were maintained below ~$7 \times 10^5$ cells per ml for up to 5 days prior to addition of brain heart infusion (BHI) broth (Sigma-Aldrich).

**Single-cell RNA sequencing.** For each scRNA-seq sample, four staggered cultures were set up over 4 days all maintained below ~$8 \times 10^5$ cells per ml during the experiment by dilution. One culture was maintained free from BHI, and the remaining had 10% BHI added 24, 48 or 72 h prior to sample preparation. Equal numbers of parasites from each culture were then combined to generate one pooled sample. 1.5 ml of the pooled culture was centrifuged, and the pelleted cells washed twice with ice-cold 1 ml 1X PBS supplemented with 1% D-glucose (PSG) and 0.04% bovine serum albumin (BSA). Cells were then resuspended in ~500 μl PSG + 0.04% BSA, filtered with 40 μm Flowmi™ Tip Strainer (Merck) and adjusted to 1000 cells/μl. In all steps, cells were centrifuged at $400 \times g$ for 10 min. In total, 15,000 cells (15 μl) from the mixed sample were loaded into the Chromium Controller (10x Genomics) to capture individual cells with unique barcoded beads. Libraries were prepared using the Chromium Single Cell 3′ GEM, Library & Gel Bead Kit v3 (10x Genomics). Sequencing was performed with the Illumina Next-Seq™ 500 platform (read one 28 bp and read two 130 bp) to a depth of ~50,000 reads per cell. Library preparation and sequencing was performed by Glasgow Polyomics. For the first WT replicate experiment, T. brucei parasites were mixed 1:1 with Leishmania mexicana prepared by the same method (Supplementary data 1), so the heterogenous doublet rate of 8.04% could be calculated.

**Read mapping and transcript counting.** The reference genome was complied with Cell Ranger v3.0.2, to combined the T. brucei TREU927 (release 50, TritrypDB) nuclear reference genome[77] and T. brucei Lister 427 maxi circle kDNA sequence (GenBank: M94286.1). 3′UTR annotations were extended to increase the proportion of reads correctly assigned to annotated transcripts. 2500 bp immediately downstream of the stop codon was assigned as the 3′UTR of each protein-coding gene, unless the existing 3′UTR was longer than 2500 UTR in which case the full length was preserved. If the new 3′UTR was overlapped with other genome features (coding and non-coding) the UTR was truncated to remove the overlap. As T. brucei samples were also multiplexed with L. mexicana parasites, a customised L. mexicana transcriptome was generated in the same way[78] and mapping was performed to both T. brucei and L. mexicana genomes. Reads were mapped and unique reads aligned to each annotated gene were counted and assigned to a cell barcode with the Cell Ranger count function (Supplementary data 1). Dual-species multiplets were also identified by Cell Ranger count. Multiplets, L. mexicana cells and all L. mexicana transcripts were subsequently removed (Supplementary data 1). Cell Ranger v3.0.2 (http://software.10xgenomics.com/single-cell/overview/welcome) was used with all default settings.

**Data processing and integration.** Count data for individual samples (WT 1, WT 2 and ZC3H20 KO) was processed separately prior to integration using the Seurat v3.2.2[79], scater v1.14.6 and Scran v1.14.6[80] packages with R v3.6.3. The percentage of transcripts encoded on the maxi circle kDNA was calculated per cell, as cells with excess proportion of mitochondrial transcripts are likely to be poor quality[37]. The percentage of transcripts per cell encoding ribosomal RNA was also calculated, as high levels of rRNA indicate poor capture of polyadenylated transcripts. Low-quality cells were removed by filtering for low total RNA (<1000), low unique transcripts (<250), high proportion of kDNA (>2%) and high proportion of rRNA (>8%). Likely doublets were removed by filtering for high total RNA (>4000) and high total unique transcripts counts (>2500). For detailed sample metrics, see Supplementary data 1.

Each filtered sample was log2 normalised individually using the quick cluster method from Scran[81]. This method employs scaling normalisation on the pooled cells firstly, to account for the high number of zero counts found in raw scRNA-seq data, before deconvolving size factors for individual cells[81]. To increase the robustness of variable genes selected for principle component (PC) analysis, we used two selection methods[82]; Scran, which uses log2 normalised transcript counts, and Seurat[79], which uses raw transcript counts. We identified 3000 genes with each method, selected those identified by both and removed VSG encoding genes[83] to avoid clustering based on VSG expression. This left 2145, 1661 and 2120 for WT 1, WT 2 and ZC3H20 KO samples, respectively (Supplementary data 1).

For integration and batch-correction of WT replicate samples, the Seurat v3 package was used[79]. Common variable features and integration anchors were identified, data for all genes integrated and scaled before the PCs were calculated using the common variable features. The first 8 PC dimensions each contributed >0.1% of additional variance and were used to select anchors and integrate data. The effect of total RNA per cell was regressed when scaling data. The ZC3H20 KO

cells were subsequently integrated with the previously integrated WT data using the steps described above, however, STACAS v1.0.1 was used to identify integration anchors as the package is specialist for samples which do not fully overlap[84].

**Cluster analysis and mark gene identification**. For clustering and marker gene analysis the Seurat v3 package was used[79]. Cells were plotted as dimensionality reduced UMAPs[38] and nearest neighbours were identified using eight dimensions. A range of clustering resolutions was trialled, with 0.4 resulting in the highest resolution clustering with significant mark genes identified for every cluster. Marker genes were identified for each cluster using MAST v1.12.0[85], a differential expression analysis tool optimised for scRNA-seq to ensure a low type-I error rate. We restricted the discovery of differentially expressed marker genes to only those expressed in >25% of the cells in the cluster, with a log2FC of >0.25 and adjusted $p$-value < 0.05. Gene ontology (GO) terms concerning biological processes were identified via the TriTrypDB[86] website ($p < 0.05$) and redundant terms removed with REVIGO[87] (allowed similarity = 0.5) and manually.

**Cell-cycle scoring**. Cell-cycle marker genes for early G1, late G1, S and G2M phases have been previously identified with bulk RNA-seq analysis[56]. We selected marker genes that were detected in at least 10% of cells to label each cell with its phase. A score for each phase was defined as the average expression of each marker gene set per cell, as calculated by the Seurat v3 MetaFeature() function. The FC of each phase score over the mean phase score across all cells was then calculated. The phase with the highest FC was assigned to each cell as its most likely phase, unless all phases had a FC < 1.5 in which case they were labelled as "non-cycling".

**Trajectory inference and pseudotime analysis**. For trajectory inference cells were plotted using PhateR v1.0.7[53] (using the same common viable genes as for PCA and eight dimensions) and trajectories were identified using slingshot[88], with the slender A cluster defined as the starting point. For cell-cycle analysis, a circular trajectory was fitted as a principle curve with princurve v2.1.4[89]. To identify genes with expression patterns associated with progression of the trajectory, generalise additive models were fit using the tradeSeq package v1.3.21[90] with default parameters. The number of knots was tested to find 6 knots provide sufficient detail for the highest number of genes without overfitting. Differential expression analysis was performed with the tradeSeq associationTest() function using default parameters. TradeSeq uses Wald tests to assess differential expression across a trajectory lineage. associationTest() tests the null hypothesis that all smoother coefficients are equal across the trajectory and computes a $p$-value for each gene based on chi-squared asymptotic null distribution of the Wald statistics[90]. Genes with $p$-value < 0.05 and mean FC > 2 were clustered using tradeSeq clusterExpressionPattern over 100 points on the trajectory. Gene clusters were merged into co-expressed modules using default setting except the merging cut-off was set to 0.95 to refine the number of modules from 58 to 9.

For early differentially expressed used earlyDETest() from TradeSeq was used, which compares expression patterns of genes across two trajectories (here WT and ZC3H20 KO differentiation trajectorys) between two fixed points (here dots shown in Fig. 6a). This test also uses Wald tests and tests the null hypothesis that a gene has the same expression pattern across both trajectories between the two points[90].

For comparison with bulk RNA-seq analysis, the fold change of stumpy vs slender expression for all genes was taken from data published by Silvester et al.[12]. All genes not found to be significant ($p$-value > 0.05) in bulk analysis were given a fold-change value of 0 for comparison with scRNA-seq data.

**Immunofluorescence**. Parasites were fixed in 1% paraformaldehyde for 10 min at room temperature (RT). Parasites were washed in 1X PBS and adhered to slides spread with Poly-L-lysine before being permeabilised with 0.1% Igepal in 1X PBS for 3 min. Cells were then blocked with 2% BSA in 1X PBS for 45 min at RT, stained with primary antibody (anti-PAD1[35] 1:1000, EP1 procyclin [Cedar labs] 1:300) diluted in 0.2% BSA for 1 h at RT. Three washes with 1X PBS were performed before incubating with secondary Alexa Fluor 488 (ThermoFisher Scientific) in 0.2% BSA for 1 h at RT. Cells were washed a further three times before mounting with Fluoromount G with DAPI (Cambridge Bioscience, Southern Biotech). Imaging was performed with an Axioscope 2 fluorescence microscope (Zeiss) and a Zeiss Plan Apochromat 63x/1.40 oil objective. Image analysis was carried out with Fiji ImageJ v2.

**Reporting summary**. Further information on research design is available in the Nature Research Reporting Summary linked to this article.

## Data availability

The transcriptome data generated in this study have been deposited in the European Nucleotide Archive with accession number PRJEB41744. The processed transcript count data and cell metadata generated in this study are available at Zenodo (https://zenodo.org/record/5163554#.YQvu2ZNKjUo)[91]. Data can be sourced via Supplementary Data Tables and wild-type scRNA-seq data can be explored using the interactive cell atlas (http://cellatlas.mvls.gla.ac.uk/TbruceiBSF/). Source data are provided with this paper.

## Code availability

Code used to perform analysis described can be accessed at Zenodo (https://zenodo.org/record/5163554#.YQvu2ZNKjUo)[91].

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

## Acknowledgements

We thank J. Galbraith and P. Herzyk (Glasgow Polyomics, University of Glasgow) for their guidance, library preparation and sequencing. We also thank M. Cayla (University of Edinburgh) for their guidance and provision of cell lines, and F.S.L. Warren for the provision of *Leishmania mexicana* samples. This research was funded in whole, or in part, by the Wellcome Trust [Grant numbers 218648/Z/19/Z to E.M.B., 104111/Z/14/ZR to T.D.O. and 103740/Z14/Z to K.R.M.]. For the purpose of open access, the author has applied a CC BY public copyright licence to any Author Accepted Manuscript version arising from this submission. This work was also Wellcome Trust Institutional Strategic Support Fund (ISSF3) awards held at the University of Glasgow (204820/Z/16/Z awarded to E.M.B. and R.M.), and the BBSRC-FAPESP (BB/N016165/1 to R.M.).

## Author contributions

Methodology: E.M.B., F.R., R.M., T.D.O. and K.R.M. Data collection: E.M.B. and F.R. Bioinformatic data analysis: E.M.B. and T.D.O. The single-cell atlas was created by T.D.O. All authors participated in discussions related to this work. All authors wrote, reviewed and approved the manuscript.

## Competing interests

The authors declare no competing interests.
