## [Peer Review File · Nature Communications]

Single-cell transcriptomic analysis of bloodstream

Trypanosoma brucei reconstructs cell cycle progression and developmental quorum sensingEditorial Note: This manuscript has been previously reviewed at another journal that is not operating a transparent peer review scheme. This document only contains reviewer comments and rebuttal letters for versions considered at Nature Communications.

Reviewers' Comments:

Reviewer #1:

Remarks to the Author:

This is a revised version of a manuscript that I originally reviewed for Nature Microbiology. The authors fixed the small stuff, but have not addressed my main concerns - the lack of separate datasets from different time points and correlation with morphology. There is clearly a reluctance to provide additional sequence data of any sort. I find this disappointing.

In response to the authors' rebuttal:

I am aware that differentiation is asynchronous, but the population at the beginning will be largely slender and the population at the end largely stumpy. The question is to what extent the stumpy population, as identified by single cell analysis, correlates with stumpy morphology. Does the stumpy transcriptome precede, coincide with or come after morphological changes? This may help resolve apparent discrepancies about whether or not slender forms are capable of efficiently infecting flies as these may already be committed to stumpiness, but not showing it.

It is true that reviewer 3 only asked for a single time point, but his/her intention was to compare differentiation in vivo and in vitro.

Reviewer #3:

Remarks to the Author:

The authors have clarified the results and added new experiments. There is still a point of view that the main element of a single cell transcriptomics approach would be the analysis of a single population containing cells at different points of the differentiation ie overcoming induction and time windows etc. However, the authors make cogent arguments why their approach is useful at this time.

Likewise with the business of an intermediate form. I would not have expected such a form to be defined by a large short of genes and still feel that small cohort/pairwise analyses of the data set might reveal more. However, again, these are views and these authors have tested some concepts. That is important and whatever I or one of other referees feel they have brought new insight to a difficult but important problem.

In summary, therefore, I believe this is a good response and the paper should be published.

Reviewer #4:

Remarks to the Author:

In the presented manuscript, Briggs et al have performed single cell RNA sequencing on bloodstream stage of the African sleeping sickness parasite *Trypanosoma brucei*. Specifically, authors have used a oligopeptide based differentiation method to essentially induce the formation of the transmissible stumpy forms from the replicative slender forms and in the process have captured and sequenced all the stages of the parasite transitioning in between the two stages using the 10X Genomics platform. From biological point of view, the authors aimed to find a discrete "intermediate" transitioning stage of

the parasite during the differentiation process. However, an “intermediate” transitioning stage was not detected in the dataset presented, leaving me to wonder what is the ground-breaking finding/achievement of the mammoth dataset presented here and how much does this add to the existing literature on *T. brucei* transcriptome. Thereafter, the authors studied gene expression differences between the slender and stumpy forms (done previously by bulk RNA sequencing by Naguleswaran et al, BMC Genomics, 2018), differential gene expression along the pseudotime trajectory and assigned cell cycle phase to each cell during the differentiation process. Lastly, the authors looked at the diverging trajectory of WT versus ZC3H20 parasites and found novel putative ‘early’ regulators of commitment. In my opinion this is the more interesting part of this work and an in-depth study and validation of differentiation regulators as well as direct / indirect targets of ZC3H20 would make a biologically more interesting study.

Comments:

- 1) Since stumpy forms can be artificially induced in vitro, could they have been purified to bypass the asynchrony in the differentiation process being an obstacle in studying gene expression changes during the development?
- 2) Authors mention in line 113-115 that bloodstream form differentiation can be induced in a titratable manner using the BHI broth. Could a bulk RNA sequencing approach have been used to study differentiation process and transitioning stages of the parasite (in a temporal manner) instead of single cell RNAseq? What was the unique advantage of applying single cell RNAseq approach here?
- 3) Regarding the technical aspects of the methodology, how do the authors remove any batch effects from the two biological replicates used in the analysis? Also, are there any controls (such as spike-ins) used during cell capturing, lysis, reverse transcription and amplification steps? If not, then how do the authors differentiate between true biological expression changes and technical noise/variation in the dataset?
- 4) It seems only 62% of the total transcriptomes captured (i.e. 9344 out of 15000 total cells) passed QC. The QC parameters discussed in line 127-140 do not account for all of the rest 38% transcriptomes rejected. Could the authors please explain on what basis the rest of the transcriptomes were removed? Also, why only cells with Feature Count of <2500 and UMI < 4000 were selected (Fig S2)?
- 5) The authors count any gene as detected if it is at least expressed in more than 10 cells (which is 0.1% of 9344 cells) (line 142-143). Is this an arbitrary cut-off? In my opinion transcripts detected in 0.1% of the population are still very rare transcripts. In that case one may instead consider all transcript as detected even if it is present in a single cell out of 9344. In the Vigneron et al, PNAS (2020) article, genes present in less than 5% of cells were filtered out. Could the authors please explain the rationale behind 0.1% cut-off in the presented study?
- 6) The median genes detected in the two replicates are 1052 and 1445 respectively (line 144). What is the reason behind this difference in transcript detection in the two replicates? Also 1052 and 1445 present ~11% and ~16% respectively of the total number of genes in *T. brucei*. Is this comparable to the transcriptome detection of mammalian cells using the 10X Genomics? Also as the authors cite reference no. 32 which show that SMARTseq2 detects 1572 genes per *T. brucei* cell, would this have been a better method of choice (especially when comparing to 1052 genes detected in replicate 1). There are also more sensitive methods recently published, such a SMARTseq3 and MALBAC-DT, have the authors considered to use these to have a better transcriptome detection?

7) PAD2, the marker gene for stumpy forms is also expressed in the slender forms (albeit at a slightly lower level) in Figure 1b. Is this expected? Also the gene expression is shown as raw transcript count in Fig 1b and it looks rather low (scale of 0-4). Is it expected that even house-keeping genes like GAPDH are expressed at such low levels in the cells? Or could this be a detection/sensitivity issue of the method used?

8) How many genes are expected to be differentially expressed between the stumpy and slender forms when sequenced by bulk RNA seq? Is this comparable to the 398 genes identified by scRNAseq of the two stages? How is the overall correlation of the single cell transcriptomes with bulk RNA sequencing data from each stage?

9) What does relative expression in Fig 1e refer to? What is it relative to? Please clarify. Also, it may give more biological insight if all 516 markers profiles are shown instead of just 10. Also, the top 10 "unique markers" defined here are overlapping between LS A and LS B (GAPDH to KIFC1 in the heatmap). Even the top 3 unique markers shown in Fig 1f are not unique in every case. For eg., ZC3H11 and SCS-alpha is expressed everywhere almost similarly. How did the authors define unique markers in that case? Furthermore, were all 516 marker genes used for the GO analysis in Fig 1g? In Fig 1f legend, "X-axis" should be corrected to "Y-axis" (line 958)

We thank all the reviewers for their feedback and have below summarised the changes we have made to the manuscript in response. We noted that reviewers queried the use of scRNA-seq as opposed to population level analysis, as well as the new insights gained with this method. In addition to new data demonstrating the asynchrony of the differentiation process *in vitro* and the relationship between morphology and developmental commitment, we have modified the manuscript text to highlight both the critical importance of using scRNA-seq and reinforce the novel findings gained.

1. We have added new data analysing the morphology of both wild-type and ZC3H20 null mutant parasites in response to oligopeptide-induction of differentiation (new figure S5). This highlights that all time points contain parasites with a range of morphology, further demonstrating that scRNA-seq is the only method of analysing these heterogenous populations. This also allowed us to ascertain that ZC3H20 null mutants do not change morphology despite down regulating slender associated transcripts. This indicates that changes to the parasite shape only occurs after regulation by ZC3H20, which operates at or close to developmental commitment.

2. We have clarified that scRNA-seq is the only currently available method for analysing slender, stumpy and transitioning parasites, explaining to the reader that in our *in vitro* model, as is the case *in vivo*, populations are always mixed in terms of both the cell cycle and developmental stage (lines 111-115 and 409-411). We now also explain that enrichment or sorting methods are not applicable for the aims of our study (lines 83-85 and 411-413).

3. We have modified and shortened the abstract (lines 19-32) to meet journal requirements and to emphasise key insights gained from our study (as discussed previously by original reviewer 2 and as outlined in our previous rebuttal). We have highlighted these below for reviewer 4. Lines 89-97, 413-424 and summary figure 6 list the new insights scRNA-seq has specifically allowed us to gain, and we have now reinforced this in lines 85-89.

Below we have responded to each reviewer individually, providing point-by-point answers for reviewer 4.

REVIEWER COMMENTS

Reviewer #1 (Remarks to the Author):

This is a revised version of a manuscript that I originally reviewed for Nature Microbiology. The authors fixed the small stuff, but have not addressed my main concerns - the lack of separate datasets from different time points and correlation with morphology. There is clearly a reluctance to provide additional sequence data of any sort. I find this disappointing.

In response to the authors' rebuttal:

I am aware that differentiation is asynchronous, but the population at the beginning will be largely slender and the population at the end largely stumpy. The question is to what extent the stumpy population, as identified by single cell analysis, correlates with stumpy morphology. Does the stumpy transcriptome precede, coincide with or come after morphological changes? This may help resolve apparent discrepancies about whether or not slender forms are capable of efficiently infecting flies as these may already be committed to

stumpiness, but not showing it.

It is true that reviewer 3 only asked for a single time point, but his/her intention was to compare differentiation *in vivo* and *in vitro*.

We thank the reviewer for revisiting this manuscript and below we have discussed the two queries raised. As regards morphology, we have generated new data to test changes in cell shape in response to BHI treatment. We have also discussed why further sequencing data would not allow us directly correlate morphology and transcriptomic changes, and nor would it change the analysis and conclusion in the original manuscript.

- In order to provide information to address the referee's question, we have compared the individual morphology of wild type (WT) cells after 72 h growth in BHI with the ZC3H20 null mutants under the same conditions and at the same time point (new figure S5). Our transcriptome data show that the ZC3H20 null mutant initiates differentiation but then fails close to commitment (Figure 5 of the manuscript). We find there is a significant increase in the width of WT parasites after 72 hr of BHI treatment, although there is considerable heterogeneity (as expected in this asynchronous differentiation). This increase in width represents the progression of slender cells to the stumpy morphology. In contrast, little change is observed in the ZC3H20 null mutant, which retains the width of cells unexposed to BHI and, hence, are slender in morphology. This finding supports the concept that morphological change is a relatively late event in the transition between slender and stumpy forms (MacGregor et al., *Cell Host Microbe*, PMID: 21501830; MacGregor et al., *Nature Reviews Microbiology*, PMID: 22543519). These new data are presented in supplementary figure S5 and we have added text relating to this in lines 313-315, and in summary figure 6.
- As previously discussed, if we performed scRNA-seq on each time point after BHI treatment, the asynchronous nature of differentiation would simply reveal the transcriptomes of a mixed population of cells at different states of development. Although there would be mainly slender cells in the first time point and a stumpy form enrichment in the last time point, this will always be a mixed population (as demonstrated in figures S1 and S5). As there are no molecular markers that define morphology, and morphological parameters cannot be captured during the Chromium single cell process, each transcriptome could still not be attributed to cells of heterogeneous shapes. Sampling different time points would not change our analysis methodology or conclusions, but instead the data would be noisier due to the use of distinct sample processing steps. The key aim of our experiments was to ensure we captured all intermediate stages during development to allow us to infer the trajectory of differentiation, which we performed successfully (Figures 2 and 5). Cell morphology *per se* (as opposed to cell type) has no known relevance for developmental commitment or transmissibility, and hence was not our main interest. Indeed, our new morphological analysis of the ZC3H20 mutant reinforces the view that commitment to differentiation is unlikely to be linked to cell morphology.

Reviewer #3 (Remarks to the Author):

The authors have clarified the results and added new experiments. There is still a point of view that the main element of a single cell transcriptomics approach would be the analysis of a single population containing cells at different points of the differentiation ie overcoming induction and time windows etc. However, the authors make cogent arguments why their approach is useful at this time.

Likewise with the business of an intermediate form. I would not have expected such a form to be defined by a large short of genes and still feel that small cohort/pairwise analyses of the data set might reveal more. However, again, these are views and these authors have tested some concepts. That is important and whatever I or one of other referees feel they have brought new insight to a difficult but important problem.

In summary, therefore, I believe this is a good response and the paper should be published.

We would like to thank the reviewer for their response and support for this manuscript. We agree (and as we rehearsed in our original rebuttal), an intermediate form could be defined by the expression of a small number of genes (potentially those found to have transient gene expression changes in this study, Figure 2d). However, it is clear that there is not a large and discrete gene expression profile characteristic of a distinct 'intermediate' development type. We have modified the manuscript text accordingly to acknowledge this possibility (lines 435-441).

Reviewer #4 (Remarks to the Author):

In the presented manuscript, Briggs et al have performed single cell RNA sequencing on bloodstream stage of the African sleeping sickness parasite *Trypanosoma brucei*. Specifically, authors have used a oligopeptide based differentiation method to essentially induce the formation of the transmissible stumpy forms from the replicative slender forms and in the process have captured and sequenced all the stages of the parasite transitioning in between the two stages using the 10X Genomics platform. From biological point of view, the authors aimed to find a discrete "intermediate" transitioning stage of the parasite during the differentiation process. However, an "intermediate" transitioning stage was not detected in the dataset presented, leaving me to wonder what is the ground-breaking finding/achievement of the mammoth dataset presented here and how much does this add to the existing literature on *T. brucei* transcriptome. Thereafter, the authors studied gene expression differences between the slender and stumpy forms (done previously by bulk RNA sequencing by Naguleswaran et al, BMC Genomics, 2018), differential gene expression along the pseudotime trajectory and assigned cell cycle phase to each cell during the differentiation process. Lastly, the authors looked at the diverging trajectory of WT versus ZC3H20 parasites and found novel putative 'early' regulators of commitment. In my opinion this is the more interesting part of this work and an in-depth study and validation of differentiation regulators as well as direct / indirect targets of ZC3H20 would make a biologically more interesting study.

We thank the reviewer for their feedback and interest in our findings. We agree that the putative early commitment regulators identified here are very interesting and agree that these require in-depth future investigation, although this is beyond what could be reasonably achieved in the current study. There are several new findings that emerge from our study of particular biological interest, which we have listed below.

1. As the reviewer acknowledges, we find the notable absence of a discrete “intermediate” transcriptome. Instead, we reveal the progressive but direct transition from a slender- to a stumpy-like transcriptome. The absence of a discrete intermediate form defined at the transcriptome level is as biologically interesting as the presence of one, particularly since intermediate forms are often referred to in the literature but are undefined in any sense. Our scRNA-seq analysis has made such cells analysable for the first time, studies that are not possible using average transcriptome studies of bulk RNA-seq on mixed populations of asynchronously differentiating parasites.
2. We further demonstrate that stumpy forms exit the cell cycle specifically before expression of late G1 transcripts and enter a G0 phase, as opposed to simply pausing in G1. This finding supports and clarifies the relationship between cell cycle position and developmental competence previously proposed by cytological studies (Matthews and Gull, *J Cell Biol*, 1994, PMID: 8195296).
3. We identify dynamic gene expression patterns across the slender cell cycle, and during slender to stumpy differentiation. With respect to asynchronous differentiation, such dynamic changes could not be identified previously, since bulk RNA-seq data simply represents a population average. This is in contrast to previous bulk RNA-seq data examining the cell cycle, where synchronous progression of a population could be achieved with centrifugal elutriation (Archer *et al.* 2011 *PLoS One* PMID: 21483801). The agreement between marker expression in cell-cycle synchronised cells and our asynchronous slender populations validates the value of our trajectory mapping strategy for mixed cell populations.
4. We have identified transcripts with transient expression patterns, some of which peak at the slender to stumpy transition point and likely include key regulatory/signalling genes. Again, these cannot be resolved in the average transcriptomes derived from bulk RNA-seq studies and identify important targets for functional follow up.
5. We validated the role of one these genes, *ZC3H20*, in the differentiation process, providing a paradigm for the dissection of regulatory control points in a non-synchronised parasite population.
6. We provide a reference cell atlas of bloodstream form differentiation from which genes differentially expressed during slender to stumpy development can be extracted. We provide a web tool for the wider community to explore these data without the need for specific bioinformatic skills.
7. We have provided an experimental framework for the deconstruction of a complex developmental pathway using a combination of single cell analysis and reverse genetic blockade.

We have modified the abstract (lines 19-32) and main manuscript text to help the reader identify these novel insights (lines 85-89), which could not have been gained by population-based methods (lines 83-85, 111-114, and 409-413).

Comments:

1) Since stumpy forms can be artificially induced *in vitro*, could they have been purified to bypass the asynchrony in the differentiation process being an obstacle in studying gene expression changes during the development?

Stumpy cells could potentially have been purified, but this would not have been informative with respect to the developmental transition between slender and stumpy forms. Both of these extremes (slender and stumpy) can be enriched *in vitro* or *in vivo* such that bulk RNA-seq can

define their transcriptome, as has been done in several studies. However, where the scRNA-seq approach has unique value is in the characterisation of the developmental steps between these cell type extremes. As highlighted above, scRNA-seq allows the reconstruction of the changes in transcriptome along the developmental pathway and allows dynamic and transient expression profiles for particular genes to be identified. These are the most interesting changes because they are likely to control particular steps or processes in the pathway. The value of studying such transcripts is highlighted by our analysis of the ZC3H20 null mutant line.

2) Authors mention in line 113-115 that bloodstream form differentiation can be induced in a titratable manner using the BHI broth. Could a bulk RNA sequencing approach have been used to study differentiation process and transitioning stages of the parasite (in a temporal manner) instead of single cell RNAseq? What was the unique advantage of applying single cell RNAseq approach here?

Titration of BHI was performed previously (Rojas *et al.* 2019 Cell PMID: 30503212). After 48 hr of treatment with varying concentrations of BHI, more PAD1 positive cells and reduced growth rate was documented as BHI concentration was increased. However, in all cases differentiation progressed asynchronously from slender to stumpy forms - simply the efficiency of differentiation changed. Again, bulk RNA-seq would only reveal the average expression of a mixed population and could not allow inference of the order of changes or the identification of dynamic expression events. We have added text in order to explain this important point in lines 111-114.

3) Regarding the technical aspects of the methodology, how do the authors remove any batch effects from the two biological replicates used in the analysis?

The data integration method employed mitigates batch effects in the data (Seurat v3, PMID: 31178118). We show in Figure S3 the result of this method and the high correlation between replicate experiments. We have now stated this in the methods section (line 593). As rehearsed in our original rebuttal, the degree of biological replication in our study exceeds that in any previous scRNA-seq studies applied to these or other parasites.

Also, are there any controls (such as spike-ins) used during cell capturing, lysis, reverse transcription and amplification steps?

Spike-in controls are not recommended by 10X genomics (<https://kb.10xgenomics.com/hc/en-us/articles/217263926-Can-ERCC-spike-ins-be-used-for-normalization->), as they often do not match the GC content, length and polyA tail length of endogenous transcripts. They are also not easily incorporated into droplet-based methods, as equal amounts are difficult to accurately incorporate into each droplet, and so cannot be used to estimate the effect of poor cell lysis or normalise expression levels. Moreover, spike-in transcripts will enter all droplets and vastly increase the amount of sequencing required to capture the endogenous transcripts and true biological variation in the data. Other droplet based scRNA-seq studies of protozoan parasites did not rely upon spike-in controls [including *Plasmodium* species (Howick *et al.* 2019 Science, PMID: 31439762), *Toxoplasma gondii* (Waldman *et al.* 2020 Cell PMID: 31955846), and *T. brucei* (Vigneron *et al.* 2020 PNAS; PMID: 31964820)].

If not, then how to the authors differentiate between true biological expression changes and technical noise/variation in the dataset?

Noise was controlled for via normalisation, scaling and principal component selection. Raw feature expression counts were normalised via a pooling method, which has been shown to improve normalisation for experiments lacking spike-ins (Aaron *et al.* 2016 Genome Biol, PMID: 27122128). Counts for multiple cells are summed, to reduce the effect of zeros within the data, and scaling normalisation performed on these pools. The pool size factors are then deconvolved to find the size factor of each cell.

A scaling step was then performed to generate z-scores (the mean expression across cells is shifted to 0 and the variance is shifted to 1). This prevents the most highly expressed genes dominating analysis. During this step we also regressed the effect of the total transcripts detected per cell, thus removing variability due to cell lysis and transcript capture efficiencies.

Prior to UMAP and PHATE dimension reduction, we calculated the variation contributed by each of the first 50 principal components (PCs). We applied a cut-off so that only the PCs which contributed an additional 0.1% of variation were included, thus removing noise (lines 596-598). PC analysis was conducted on the top robust variant genes: lines 586-591, supplementary data 1.

We have added a brief explanation to the methods sections (lines 584-586) and all annotated code has been made available (https://github.com/emma23ed/Tbrucei_scRNA-seq). Importantly, we validated our findings by investigating the expression of genes known to be variable in the differentiation process (Figures 1a, 2c and S3c), compared to bulk RNA-seq data (Figures 2e and S3c) and performed experimental validation of ZC3H20 KO (Figures 4 and 5).

4) It seems only 62% of the total transcriptomes captured (i.e. 9344 out of 15000 total cells) passed QC. The QC parameters discussed in line 127-140 do not account for all of the rest 38% transcriptomes rejected.

This is a misunderstanding, which we have now clarified in the text (lines 134-136). Approximately 15,000 cells were used at the very start of each experiment and, naturally, not all cells are captured. Prior to quality control filtering, 7,209 and 4,263 cells were identified for the two WT replicates. Hence, our raw recovery rate is 48% and 28%, respectively. These figures are included in supplementary data 1, and this information is provided in the text (line 138).

Chromium recovery rates are variable, depend on cell type and between experiments. 10x genomics cites up to 65% as a high recovery rate for human cells.

Could the authors please explain on what basis the rest of the transcriptomes were removed? Also, why only cells with Feature Count of <2500 and UMI < 4000 were selected (Fig S2)?

We removed cells with feature count of >2,500 and UMI >4,000, as cells above these thresholds contain far above the average RNA content (Figure S2) and are likely to have originated from multiple cells captured within the same droplet. We removed further cells with significantly lower than average features and UMI, as these are likely to be of lower

quality, perhaps due to poor cell lysis or transcript capture. These are key steps for the quality control of all scRNA-seq data sets. We additionally removed cells with high kDNA (mitochondrial genome) content, as these are also likely to be poor quality: in lysed cells cytoplasmic transcripts are lost at a higher rate than mitochondrial transcripts, and so those with high proportions of mitochondrial transcripts are likely to have lysed prior to cell capture and lysis (Ilicic *et al.* 2016 Genome Biol., PMID: 26887813). Lastly, non-polyadenylated rRNA transcripts should not be captured in our poly(T) capture method, and so cells with high rRNA are also low quality and removed. These thresholds are explained in lines 573-581 and metrics are available in supplementary data 1.

5) The authors count any gene as detected if it is at least expressed in more than 10 cells (which is 0.1% of 9344 cells) (line 142-143). Is this an arbitrary cut-off? In my opinion transcripts detected in 0.1% of the population are still very rare transcripts. In that case one may instead consider all transcript as detected even if it is present in a single cell out of 9344. In the Vigneron et al, PNAS (2020) article, genes present in less than 5% of cells were filtered out. Could the authors please explain the rationale behind 0.1% cut-off in the presented study?

We have clarified in the text (line 138 and line 147) that these genes are only retained in the data set for completeness and do not contribute to the variable genes selected. This means they are not considered when calculating PCs and, thus, do not influence clustering or trajectory analysis. This will also be the case for the Vigneron data set, where the most variable genes were also selected. We have provided the variable genes used in supplementary data 1, where we have now included the proportion of cells in which each gene is detected (averaging 17.72%).

When identifying cluster marker genes, we applied a cut-off so each gene has to be present in 25% of cells in the cluster to be identified as a significantly differentially expressed.

6) The median genes detected in the two replicates are 1052 and 1445 respectively (line 144). What is the reason behind this difference in transcript detection in the two replicates? Also 1052 and 1445 present ~11% and ~16% respectively of the total number of genes in *T. brucei*. Is this comparable to the transcriptome detection of mammalian cells using the 10X Genomics?

The number of genes detected in single cell experiments varies due to cell quality, efficiency of droplet formation, cell lysis, reverse transcription and cDNA amplification. It is not possible from the data to conclude the cause of the variability, although it is unlikely to be due to cell quality, as all samples contained > 90% viable cells. The number of genes also varies with cell type, and as the *T. brucei* genome is far smaller (~100x) than the human and mouse genomes, as is RNA content, we would not expect the same level of recovery. As discussed on lines 140-145, our data compare very favourable to other studies of *T. brucei*, as well as with other protozoan parasites (PMID: 33678213).

Also, as the authors cite reference no. 32 which show that SMARTseq2 detects 1572 genes per *T. brucei* cell, would this have been a better method of choice (especially when comparing to 1052 genes detected in replicate 1). There are also more sensitive methods

recently published, such as SMARTseq3 and MALBAC-DT, have the authors considered to use these to have a better transcriptome detection?

SMART-seq and MALBAC-DT are plate-based methods and so cells must be separated by placing individual cells into separate PCR tubes or plate wells. This means the number of cells that can be analysed is restricted compared with Chromium from 10x Genomics. As we did not know how prevalent any rare cell populations would be in our experiments, we chose this method to capture thousands of cells and to avoid missing any key parasite forms. High cell numbers are also preferable for trajectory analysis, which was one of our key aims, in order to deconvolve the differentiation process.

7) PAD2, the marker gene for stumpy forms is also expressed in the slender forms (albeit at a slightly lower level) in Figure 1b. Is this expected?

Yes, low levels of PAD2 mRNA are detected in slender forms by blotting, and are increased in stumpy forms (Dean *et al.* 2009 *Nature*, PMID: 19444208).

Also the gene expression is shown as raw transcript count in Fig 1b and it looks rather low (scale of 0-4). Is it expected that even house-keeping genes like GAPDH are expressed at such low levels in the cells? Or could this be a detection/sensitivity issue of the method used?

Low raw transcript counts are expected in single cell transcriptomics, as the aim is to label and capture transcripts prior to cDNA amplification. This is a technical challenge common to all scRNA-seq methods.

8) How many genes are expected to be differentially expressed between the stumpy and slender forms when sequenced by bulk RNA seq? Is this comparable to the 398 genes identified by scRNAseq of the two stages? How is the overall correlation of the single cell transcriptomes with bulk RNA sequencing data from each stage?

We have performed this correlation analysis in supplementary figure S3b, for both WT replicates, and find a good correlation ($R = 0.777$ and 0.756). The number of differentially expressed genes in bulk RNA-seq from slender and stumpy, *in vivo* derived populations, with a fold-change >2 , is 923 (PMID: 30307943). Simply comparing slender and stumpy clusters in our scRNA-seq data reveals fewer genes, likely due to the lower transcript detection rate compared to bulk RNA-seq. However, the advantage of scRNA-seq is the ability to perform differential expression analysis across the differentiation process, instead of comparing populations. This revealed 1,791 genes that were differentially expressed, compared with 1,222 genes (68%) identified in bulk RNA-seq analysis. This comparison is discussed on lines 217-225 and presented in figure 2e.

9) What does relative expression in Fig 1e refer to? What is it relative to? Please clarify.

Relative expression is the scaled z-scores (explained above) for this set of genes, required for heatmap plotting. We have now changed the figure legends accordingly.

Also, it may give more biological insight if all 516 markers profiles are shown instead of just 10.

All markers are detailed in supplementary data 2; unfortunately, it is difficult to clearly plot all markers with labels on the same heatmap. Instead, we gain insight from these markers by highlighting the associated GO terms (Fig 1g).

Also, the top 10 “unique markers” defined here are overlapping between LS A and LS B (GAPDH to KIFC1 in the heatmap). Even the top 3 unique markers shown in Fig 1f are not unique in every case. For eg., ZC3H11 and SCS-alpha is expressed everywhere almost similarly. How did the authors define unique markers in that case?

Unique is defined as significantly increased in expression in the cluster in question, compared to all other clusters. For example, a gene found to be expressed at the same level in both LS A and LS B, but increased relative to the SS A and SS B clusters, would not be considered a marker gene. However, we appreciate this isn't clear and have removed this description from the legend and text.

Furthermore, were all 516 marker genes used for the GO analysis in Fig 1g?

Yes, for each cluster all markers were used, excluding those encoded by the kDNA (mitochondrial genome) as no GO terms are assigned to these genes in the available databases.

In Fig 1f legend, “X-axis” should be corrected to “Y-axis” (line 958)

This has been corrected

Reviewers' Comments:

Reviewer #4:

Remarks to the Author:

The authors have clarified most of the questions, however some of my queries still remain unanswered. My comments to the authors are listed below.

The authors state "We further demonstrate that stumpy forms exit the cell cycle specifically before expression of late G1 transcripts and enter a G0 phase" in the rebuttal. Please clarify which figure is demonstrating this exactly? This has not been discussed anywhere in the relevant results section in line 254 to 294. It is also not clear from Fig 3a and Fig 4i that stumpy forms are in G0 phase. If anything, the stumpy forms seem to have a mixed distribution of various cell cycle phases (as mentioned by authors in line 263-266). There is no Fig 4j shown as mentioned in line 436. Please clarify how this conclusion was derived.

Point 4) of the rebuttal: It is reasonable to suspect that transcriptomes showing high feature count and UMI may consist of multiple cells, however, the distribution in Fig S2a looks continuous. What is the red dotted lines (cut-offs) based upon? How the cut-offs were decided for UMI count and feature count? Is this based on indications from any bulk transcriptome controls (such as 1000 cells) etc.?

Point 7) of the rebuttal: Low transcript detection is specially an issue with 10X Chromium platform. I understand that the authors aimed to capture large amount of cells in order to detect rare cells type (and hence didn't use plate-based methods), nonetheless, it is important to clearly state that low raw read counts may be due to sensitivity issue. Even when labelling the transcripts prior to amplification, do we expect such low raw transcript numbers for housekeeping gene? If this is not the case, then the authors should comment on possible transcript detection sensitivity issue when using Chromium.

Point 8) of the rebuttal: To clarify, my question is regarding whole transcriptome expression correlation between single cell and bulk transcriptome. Fig S3b shows correlation of logFC versus log2FC of only differentially expressed genes. Could the authors comment on overall transcriptome correlations? This will better reveal the quality of the overall data.

REVIEWER COMMENTS

We would like to thank reviewer #4 for revisiting the manuscript and providing additional feedback. The cell cycle status of stumpy cells had also puzzled us and so we have refined the cell cycle scoring method and added additional analysis (new figure 3), which reveals cell cycle exit specifically into a non-cycling phase for the majority of stumpy parasites during differentiation. We feel this new analysis has improved the manuscript and hope the finding is now clearer.

We have also provided additional data for the reviewer regarding RNA levels per cell and a transcriptome-wide comparison with bulk RNA-seq data. Lastly, the reviewer brought to our attention figure S2 leading us to regenerate these graphs. In doing so we noticed the tool in use had incorrectly coloured the data points. We apologise for this oversight and have manually recreated the plots to correct this mistake. Please note, none of our downstream analysis or conclusions are affected by this.

Reviewer #4 (Remarks to the Author):

The authors have clarified most of the questions, however some of my queries still remain unanswered. My comments to the authors are listed below.

The authors state “We further demonstrate that stumpy forms exit the cell cycle specifically before expression of late G1 transcripts and enter a G0 phase” in the rebuttal. Please clarify which figure is demonstrating this exactly? This has not been discussed anywhere in the relevant results section in line 254 to 294. It is also not clear from Fig 3a and Fig 4i that stumpy forms are in G0 phase. If anything, the stumpy forms seem to have a mixed distribution of various cell cycle phases (as mentioned by authors in line 263-266). There is no Fig 4j shown as mentioned in line 436. Please clarify how this conclusion was derived.

We have improved our approach to cell cycle analysis, with two outcomes: 1) we show that most stumpy parasites do indeed exit the cell cycle; and 2) we more clearly demonstrate that stumpy forms emerge during early G1 of the slender form cell cycle. This analysis has now been separated (new figure 3) from the analysis of the slender form cell cycle (now presented in new figure 4) and is explained below.

In the previous cell cycle labelling system, the average expression of marker genes for each cell cycle phase was calculated to give a phase score per individual cell of early G1, late G1, S and G2M, based on the top scoring phase. The lack of grouping of stumpy forms by cell cycle stage was clearly different to slender forms, where each cell grouped logically by phase.

We have now improved this analysis by calculating the fold-change of each phase score (defined above) per cell over the average phase score across all cells (as performed previously by Triosh *et al. Science* (2014); PMID: 27124452). For each cell, the phase with the highest fold-change was selected, unless fold-change was below 1.5 for all phases, in which case the cells are labelled “non-cycling”. This is now explained on lines 614-623 of the methods section. This approach highlights the cells which do not overexpress any set of phase markers, and so are likely to be non-cycling (Fig. 3). As with our previous analysis, cycling slender cells group clearly according to phase (Fig. 3a) and now the enrichment for non-cycling cells in the stumpy clusters, and even in slender cells most proximal to stumpy clusters, is evident (Fig 3a. and 3b). Interestingly, some cells expressing slender-associated transcripts down-regulated all cell cycle phase marker gene sets (Fig. 3c). These may reflect cells that have exited the cell cycle independently of the differentiation programme, or cells that have exited the cell cycle but not irreversibly. Conversely, some cells expressing stumpy marker genes and clustering into stumpy A and stumpy B clusters still express genes associated with an active cell cycle

phase, particularly early G1 (Fig 3c). This is intriguing but may reflect the relative dynamics of mRNA abundance for cell cycle and developmental markers, or stochasticity.

In order to more clearly show that stumpy cells exit the slender cell cycle in early G1, we have added new plots 3d-f. Here we have isolated slender cells and replotted the UMAP (using the same top variable genes as for plot 3a). In figure 3d slender cells group by cell cycle phase with most non-cycling cells positioning with early G1 cells. To map the progression, we inferred a trajectory and assigned pseudotime values (Fig. 3e). The ridge plot in figure 3f shows the number of cells in each phase, or non-cycling, across pseudotime. Most cells are in late G1 at the start of pseudotime followed by S and G2M phases. Most early G1 and non-cycling cells are found at the end of the trajectory. Hence, non-cycling cells exit during early G1 and commitment to exit takes place prior to expression of late G1 transcripts.

Usefully, by excluding slender cells labelled as non-cycling, our refined analysis method also allowed us to remove noise from our analysis of gene expression during the slender cell cycle. Accordingly, we have repeated this analysis, now presented in figure 4, and have modified the manuscript to include this analysis where signal and noise are better resolved.

We have also reanalysed the cell cycle phases for the ZC3H20 KO cells (Figs 5i and 5j). Interestingly, the cluster of cells labelled LS B.2, which contains mostly ZC3H20 KO mutants, is enriched for non-cycling cells (Fig. 5i, j), in agreement with their slowed growth phenotype (Fig. 5a). This supports the view that ZC3H20 null mutants, although morphologically slender (Fig. S5), down-regulate slender transcripts and begin reversible (Fig S4) exit from the cell cycle, consistent with this regulator operating at, or close to developmental commitment. We have revised the manuscript text to reflect this new analysis (lines 342-344).

Point 4) of the rebuttal: It is reasonable to suspect that transcriptomes showing high feature count and UMI may consist of multiple cells, however, the distribution in Fig S2a looks continuous. What is the red dotted lines (cut-offs) based upon? How the cut-offs were decided for UMI count and feature count? Is this based on indications from any bulk transcriptome controls (such as 1000 cells) etc.?

Yes, the distribution is continuous in all cases and so it was necessary to decide on the cut-offs after several iterations of data analysis and inspection of the results for likely doublets. For the reviewer's information below is a hexagonal heatmap to show the distribution of cells by total UMI count (x-axis) and total feature count (y-axis).

While generating this plot we noticed an error in figure S2 in our previous manuscript submission. The tool used to generate the previous plots had incorrectly labelled the data points. We have now manually generated these plots and corrected our mistake in new figure S2, adding a missing line indicating the lower UMI cut-off. Please note, this has not impacted our analysis, only the appearance of the plots in question.

Additionally, for the reviewer's information, we have plotted the complete WT data without applying cut-offs (a-e) and compared this to our final dataset containing only filtered cells (f-j).

High resolution clustering of the unfiltered data set (a) was performed to identify clusters with high or low average total UMIs (b) and features (c). Figures d and e show cluster 2 contains the cells with highest RNA content, whereas clusters 6 and 9 contain far lower RNA content. Figures g-j show the more even distribution of cells by UMI and feature counts in our final dataset (fig f). Please note, no significant marker genes could be identified for low RNA clusters and so these are more likely to represent poor quality transcriptomes as opposed to biologically distinct cell types, although the latter cannot be completely ruled out.

Point 7) of the rebuttal: Low transcript detection is specially an issue with 10X Chromium platform. I understand that the authors aimed to capture large amount of cells in order to detect rare cells type (and hence didn't use plate-based methods), nonetheless, it is important to clearly state that low raw read counts may be due to sensitivity issue. Even when labelling the transcripts prior to amplification, do we expect such low raw transcript numbers for housekeeping gene? If this is not the case, then the authors should comment on possible transcript detection sensitivity issue when using Chromium.

House-keeping genes are difficult to define for this dataset, as many genes traditionally thought to be "house-keeping" are differentially expressed between slender and stumpy forms (including GAPDH and tubulin). Although low transcript counts are an issue for all single cell transcriptomic methods, we find our Chromium data compares well against other studies of unicellular parasites and various approaches (see table below). We have, however, added a statement referring to the issue of low sensitivity and the likelihood that genes with low expression have not been captured in our experiments (line 508-509).

Technique	Species	Cells recovered per experiment described	Average genes per cell	Reference
SMART-seq2	P. falciparum	161-191	1712-2090	(Reid et al. 2018)
	P. berghei	102-517*	202-2995*	(Howick et al. 2019)
	P. berghei	144	1981	(Reid et al. 2018)
	T. gondii	849-2198	862-1290	(Xue et al. 2020)
	T. brucei	418	1572	(Müller et al. 2018)
	P. falciparum (iRBCs)	92	300-800*	(Ngara et al. 2018)
Chromium	P. falciparum	6737	438	(Howick et al. 2019)
	P. berghei	4884	791	(Howick et al. 2019)
	P. knowlesi	4237	557	(Howick et al. 2019)
	P. vivax	22-2098	1019 ^s	(Sà et al. , 2020)
	T. brucei	2045	298	(Vigneron et al. 2020)
	T. brucei	2295-5321	1117-1494	(Briggs et al. 2020)
Drop-seq	P. falciparum (iRBCs)	436-2993	224-681	(Poran et al. 2017)
Seq-well	T. gondii	26560	685	(Waldman et al. 2020)
SCRB-Seq	P. falciparum	191-364	212-503	(Brancucci et al. 2018)
NEBNext	P. falciparum	88-451*	108-1496.5*	(Real et al. 2020)

Point 8) of the rebuttal: To clarify, my question is regarding whole transcriptome expression correlation between single cell and bulk transcriptome. Fig S3b shows correlation of logFC versus log2FC of only differentially expressed genes. Could the authors comment on overall transcriptome correlations? This will better reveal the quality of the overall data.

In fact, the requested comparison is not appropriate because the data type and source material is different; one comprises the transcriptomes of a mix of different cells types averaged as a pool, the other is a representation of individual cell transcriptomes. As a consequence, the comparisons will be valid for quite strongly differentially regulated genes only, as we presented. To emphasise this point we show below the plot of the two data types, where only the most strongly regulated transcripts correlate well, as expected. In the plot we have included all genes found in at least 1% of cells in the scRNA-seq wild-type data. Genes with adjusted p-value > 0.05 in either data set are shaded grey. As discussed above, the low sensitivity of scRNA-seq experiments means only genes in a higher percentage of cells are normally considered in differential expression analysis to avoid false positives. In figure S3, we include only genes detected in 25% of cells for this reason. Please note, kDNA encoded genes (including COII) where not included in the bulk RNA-seq analysis.

Reviewers' Comments:

Reviewer #4:

Remarks to the Author:

The authors now present new analysis on cell cycle phase assignment which shows that majority of stumpy forms are non-cycling (Fig 3a and 3b). This has raised a few minor questions as follows:

- First of all, there is a massive discrepancy in the cell cycle phase between Replicate A and Replicate B. This is especially apparent for the slender A and slender B forms, where LSA primarily consists of late G1 and S phase whereas LSB shows mixed distribution of G2M cells, non-cycling parasites and early and late G1 parasites. How do the authors explain this difference in the two slender populations? Is this expected? Or is it because the samples collected from the two bioreplicates were not in the same developmental stage? The pseudotime values in Fig 3e indicate that. Does this indicate that older slender forms become asynchronous in their development? The stumpy forms look largely consistent in the non-cycling phase with minor differences in the early and late G1 phase distribution.
- Fig 3c seems to have a highly confusing color scheme. The Cluster colors shown look exactly like the cell cycle phase color key in Fig 3b! Whereas, the cell cycle phase color scheme looks different from 3b with new light blue color introduced. So the legend "Phase key as in b" in line 1022 doesn't work. Where are the non-cycling cells shown in Fig 3c? Is it the one in blue instead of grey?? Please choose a distinctively different color scheme for the cell cycle phase so the readers don't mix up clusters and phases.
- Is the color contrast for Fig 3c really low or most genes shown are expressed at very low levels? The heatmap looks rather "blackish" and could be improved visually in my opinion.
- Fig 3d seems redundant as the information is already shown in Fig 3a, but the authors may keep it if they deem it as necessary.

REVIEWER COMMENTS

Reviewer #4 (Remarks to the Author):

The authors now present new analysis on cell cycle phase assignment which shows that majority of stumpy forms are non-cycling (Fig 3a and 3b). This has raised a few minor questions as follows:

- First of all, there is a massive discrepancy in the cell cycle phase between Replicate A and Replicate B. This is especially apparent for the slender A and slender B forms, where LSA primarily consists of late G1 and S phase where as LSB shows mixed distribution of G2M cells, non-cycling parasites and early and late G1 parasites. How do the authors explain this difference in the two slender populations? Is this expected? Or is it because the samples collected from the two bioreplicates were not in the same developmental stage? The pseudotime values in Fig 3e indicate that. Does this indicate that older slender forms become asynchronous in there development? The stumpy forms look largely consistent in the non-cycling phase with minor differences in the early and late G1 phase distribution.

This is a misunderstanding on the reviewer's part. Replicate experiments are not presented separately in Fig. 3, rather separate cell clusters (defined in Fig. 1 of the manuscript) are shown.

Specifically, slender A and slender B do not represent replicate experiments, they are the clusters of transcriptionally similar cells identified in Fig. 1 and discussed throughout the paper. These are first introduced in lines 167-175. Our biological replicate experiments are instead consistently named WT 1 and WT 2 and show very good reproducibility, as we previously highlighted but reiterate below. The cells originating from each experiment are also clearly presented in Figs. 1a, S3a and S3c. They are first discussed in lines 118-119. We also clearly state that slender B cells are more enriched for non-cycling cells, reflecting this cluster's more advanced development towards the stumpy form compared to slender A cells (lines 261-262; Fig. 3a).

In any case, below we have provided the proportion of cells in each cell cycle phase, for WT 1 and WT 2 replicate experiments. They are made up of similar proportions and reflect slight differences in growth dynamics between these two biological replicates. Clearly there is not a "massive discrepancy" between our replicate experiments, as stated incorrectly by the reviewer above. We have also added text to the manuscript to state explicitly that slender A and slender B refer to clusters identified previously, not to replicates (lines 259-260).

"Analysing the clusters presented in Fig. 1, revealed that 94.1% of slender A parasites were expressing cell cycle marker genes, whereas slender B cells included 46.6% that were labelled as non-cycling."

a) Percentage of cells from each biological replicate experiment (WT 1 and WT 2) in each cell cycle phase. **b)** Growth curves of cultures used for biological replicate experiment.

- Fig 3c seems to have a highly confusing color scheme. The Cluster colors shown look exactly like the cell cycle phase color key in Fig 3b! Whereas, the cell cycle phase color scheme looks different from 3b with new light blue color introduced. So the legend “Phase key as in b” in line 1022 doesn’t work. Where are the non-cycling cells shown in Fig 3c? Is it the one in blue instead of grey?? Please choose a distinctively different color scheme for the cell cycle phase so the readers don’t mix up clusters and phases.

We have now corrected the colour scheme in figure 3c to match the other figures.

- Is the color contrast for Fig 3c really low or most genes shown are expressed at very low levels? The heatmap looks rather “blackish” and could be improved visually in my opinion.

Yes, the colour indicates the lack of differential expression of cell cycle phase marker genes in cells we have found to be “non-cycling”. Therefore, any chosen heatmap colour scale will appear largely uniform for non-cycling cells. We would prefer to keep the colours consistent with figures 1e and S6.

- Fig 3d seems redundant as the information is already shown in Fig 3a, but the authors may keep it if they deem it as necessary.

In the previous submission, the reviewer asked us to more clearly demonstrate that cells exit the cell cycle in early G1. We therefore generated figures 3d, 3e and 3f, which focus on the early stages of cell cycle exit. We are also aware that coloured UMAP plots of single cell data can be difficult for some readers to evaluate. We hence would like to retain the ridge plot in Fig. 3f, and Figs. 3d and 3e are necessary to explain how the plot was generated.